# RECURSIVE CLEANING FOR LARGE-SCALE PROTEIN DATA VIA MULTIMODAL LEARNING

## ABSTRACT

Reliable datasets and high-performance models work together to drive significant advancements in protein representation learning in the era of Artificial Intelligence. The size of protein models and datasets has grown exponentially in recent years. However, the quality of protein knowledge and model training has suffered from the lack of accurate and efficient data annotation and cleaning methods. To address this challenge, we introduce **ProtAC**, which corrects large **Prot**ein datasets with a scalable **A**utomatic **C**leaning framework that leverages both sequence and functional information through multimodal learning. To fulfill data cleaning, we propose the Sequence-Annotation Matching (SAM) module in the model, which filters the functional annotations that are more suitable for the corresponding sequences. Our approach is a cyclic process consisting of three stages: first pretraining the model on a large noisy dataset, then finetuning the model on a small manually annotated dataset, and finally cleaning the noisy dataset using the finetuned model. Through multiple rounds of "train-finetune-clean" cycles, we observe progressive improvement in protein function prediction and sequence-annotation matching. As a result, we achieve **(1)** a state-of-the-art (SOTA) model that outperforms competitors with fewer than 100M parameters, evaluated on multiple function-related downstream tasks, and **(2)** a cleaned UniRef50 dataset containing ∼50M proteins with well-annotated functions. Performing extensive biological analysis on a cleaned protein dataset, we demonstrate that our model is able to understand the relationships between different functional annotations in proteins and that proposed functional annotation revisions are reasonable.

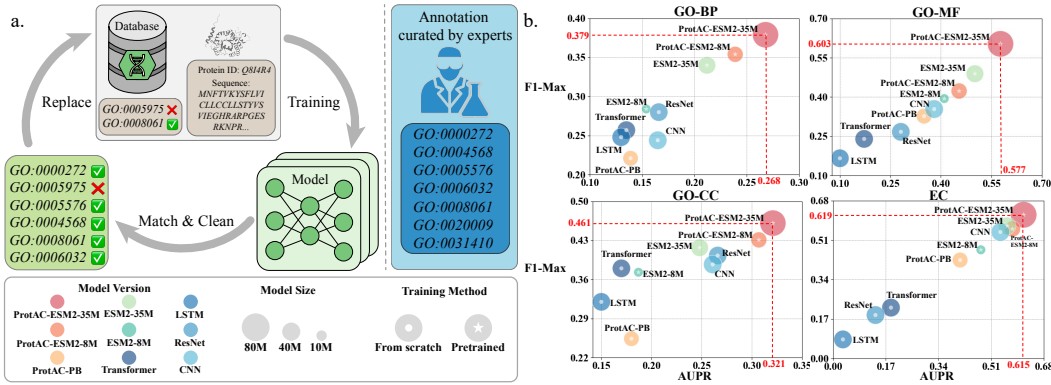

Figure 1: **(a)** Schematic diagram of recursive data cleaning for protein function annotation and expert-curated ground-truth annotations. We take protein ID Q8I4R4 as an example. This cycle is repeated many times, and the modified annotations are more consistent with the results of manual screening by biologists than the original annotations in the database. **(b)** Performance of ProtAC and other models with less than 100M parameters on downstream tasks related to function prediction.

# 1 INTRODUCTION

Proteins, central components of cellular machinery, have been the focus of extensive experimental and computational approaches aimed at elucidating their functions. The advent of high-throughput sequencing technologies (Reuter et al., 2015) has led to a significant increase in the number of sequenced genomes in past two decades, resulting in the creation of extensive protein databases. These databases serve as training resources for the advancement of deep learning in protein research (Chen et al., 2024; Elnaggar et al., 2021; Lin et al., 2023; Ferruz et al., 2022; Nijkamp et al., 2023).

Language models (LM) are highly valued for their effectiveness in natural language processing, and a specialized version known as Protein Language Model (PLM) has been extensively utilized in protein representation learning. This variant leverages protein sequences as training data, as amino acid sequences serve as the fundamental coding for proteins. PLMs demonstrate exceptional capabilities in comprehending protein functions (Rives et al., 2021; Brandes et al., 2022; Meier et al., 2021; Vig et al., 2020) and structures (Rives et al., 2021; Lin et al., 2023; Rao et al., 2020; Vig et al., 2020), thereby facilitating *de novo* protein design (Verkuil et al., 2022).

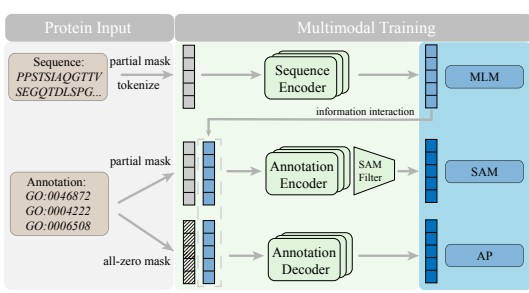

Figure 2: Model architecture and training objectives of ProtAC.

Recent studies (Xu et al., 2023; Zhang et al., 2023b) have demonstrated that PLMs leveraging multimodal information, such as sequence, functional annotation, and structure data from proteins, exhibit superior capabilities compared to models pretrained solely on sequences. However, despite significant advancements in computational methods, particularly in deep learning, which have achieved near laboratory-level precision in protein structure prediction (Jumper et al., 2021; Baek et al., 2021; Abramson et al., 2024) and expanded the structural coverage of the known protein-sequence space (Varadi et al., 2022), accurate protein function prediction remains a challenge. High-quality protein structure databases (Burley et al., 2017) and biological knowledgebases (Boutet et al., 2007) are still relatively limited in scale compared to the vast amount of validated sequences available. Existing automatic annotation methods, primarily statistical and rule-mining-based approaches (Consortium, 2019) applied to large-scale protein datasets, often face challenges when applied to large-scale protein datasets due to the complex mapping between protein sequences and functions, resulting in inaccuracies in protein property annotations. These issues[1][2] not only impact data quality but also introduce uncertainty into subsequent research endeavors (MacDougall et al., 2020; Aleksander et al., 2023). Therefore, the identification and removal of noise and errors to enhance the accuracy and reliability of protein datasets are crucial in the fields of bioinformatics and proteomics. Effective solutions are urgently needed to address these challenges.

Building upon the latest advancements in protein multimodal learning methods (Xu et al., 2023; Brandes et al., 2022), we propose an innovative learning framework that integrates multiple modalities of protein data, including sequence and functional information. Drawing inspiration from the concept of matching in Vision-Language Learning (Li et al., 2021; 2022), our framework effectively discerns between reliable and unreliable information within large-scale protein datasets. Our approach introduces a novel multi-round training strategy, where each round involves model pretraining on a noisy dataset followed by finetuning on a manually curated dataset. Subsequently, the model is tasked with cleansing the noisy dataset by predicting and selecting credible protein function information. The cleaned dataset is then recursively utilized in the next round of pretraining. A visual representation of the concept of recursive data cleaning for protein datasets is depicted in Fig.1a. This iterative process enables the replacement of noisy datasets in subsequent rounds, leading to mutual enhancement of both dataset quality and model performance.

---

[1]https://www.uniprot.org/help/evidences

[2]https://www.ebi.ac.uk/QuickGO/term/ECO:0007669

Our study explores efficient model training approaches and finds that using pretrained weights is more effective than training from scratch for enhancing model training and dataset quality. Models with larger parameter sizes outperform smaller models in functional prediction tasks and dataset quality improvement. Our model achieves SOTA results in protein function prediction tasks, surpassing models with similar parameter sizes (Fig.1b) and remaining competitive against larger PLMs. We evaluate the data cleaning capabilities of our model using the newly updated SwissProt dataset, which experts have validated but the model has never seen. This evaluation shows that our model has exceptional anomaly detection capabilities and greatly improves the accuracy of protein function prediction, nearly matching the proficiency of human biologists. Furthermore, we conduct a detailed biological analysis of the cleaned dataset, successfully verifying that our modifications to noisy protein information are biologically meaningful and align with biological principles.

## 2 RELATED WORK

**Protein Multimodal Learning**    Mutual understanding of sequence and function plays a significant role in exploring biological behaviors. Recently, multimodal models have been developed to integrate information from protein sequence and function. ProteinBERT (Brandes et al., 2022) adopts the classical BERT architecture and leverages local attention to integrate protein sequence information and utilizes global attention to learn function information; OntoProtein (Zhang et al., 2022) learns protein representations under the context of a knowledge graph, which contains GO text description and related protein information; ProGen (Madani et al., 2020) incorporates protein function labels to generate functional proteins, but it lacks the consideration the role that biomedical text can play. ProtST (Xu et al., 2023) enhances both representation learned by protein sequence and biomedical texts.

**Protein Functional Annotation Prediction**    The accuracy of protein function prediction is an important reflection of PLM capabilities. Gene Ontology (GO) (Ashburner et al., 2000) annotations provide a detailed description of protein functions in biological systems. Predicting GO annotations for uncharacterized proteins is crucial for exploring unknown protein landscapes. In each Critical Assessment of Functional Annotation (CAFA), several noteworthy protein GO annotation prediction models appear (Yao et al., 2021; Wang et al., 2023; Kulmanov et al., 2018; Zhou et al., 2019), showing significant progress. AnnoPRO (Zheng et al., 2023) combines protein sequence representation with GO functional family information to capture the intrinsic correlation between protein features and significantly improve the annotation performance of low-abundance protein families.

**Knowledge Distillation and Data Cleaning**    Knowledge Distillation (KD) (Hinton et al., 2015) aims to improve the performance of student models by distilling knowledge from teacher models. Different from most existing KD methods, which simply force the student to have the same category predictions as the teacher, Li et al. (2022) proposed CapFilt, which can be interpreted as a more effective way to perform KD in the context of Vision-Language Pretraining (VLP), where the captioner distills knowledge through semantically rich synthetic captions, while the filter distills knowledge by removing noisy captions. We apply this idea to large protein dataset cleaning for the first time. See more related work in Appendix A.

## 3 METHOD

We present ProtAC, an automatic data cleaning framework for large protein datasets that leverages unified knowledge from protein sequence and functional annotations. In this section, we first introduce our model architecture and its training objectives, and then describe our data cleaning strategy.

### 3.1 MODEL ARCHITECTURE

**Overview:**    Our model structure consists of four main components: Sequence Encoder, Annotation Encoder, Annotation Decoder, and SAM Filter (the model architecture is shown in Fig.2). As a versatile learning framework, the first three main modules can be easily replaced with mainstream PLMs, which greatly expands the scope of further model design and lays the foundation for inspiring future directions of protein multimodal representation learning.

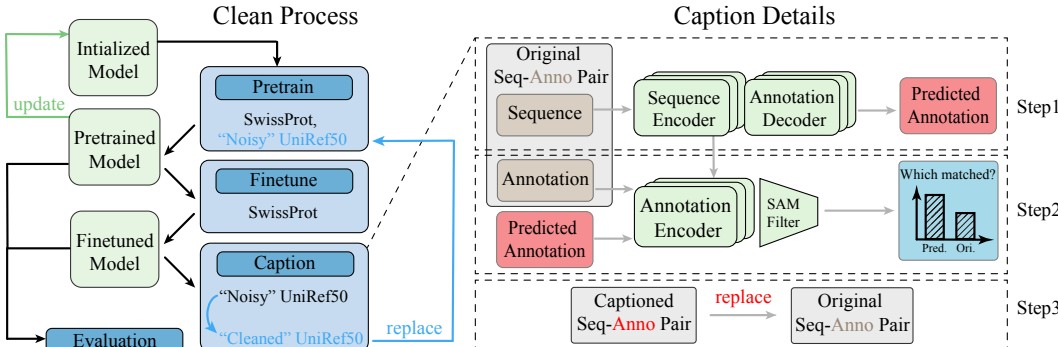

Figure 3: Data cleaning workflow of ProtAC. The left part of the figure outlines the cleaning process, and the right half of the figure details the caption process.

**Sequence Encoder:** We investigate two widely used PLMs: ESM2 (Lin et al., 2023), one of the current SOTA sequence-only PLMs, known for its superior protein feature extraction capabilities, making it a common choice for protein representation learning, especially in multimodal learning tasks; ProteinBERT (Brandes et al., 2022), a multimodal PLM based on the BERT architecture, which captures sequence information through its local part and functional information through its global part, with a cross-attention mechanism promoting the interaction between the two parts. For our Sequence Encoder, we use the local part of ProteinBERT and ESM2. Input sequence is partially masked and tokenized.

**Annotation Encoder and Decoder:** We modify the global part of ProteinBERT as our Annotation Encoder. The input annotation information in the form of a fixed-size binary vector is partially masked and encoded into the annotation embedding. The sequence information is injected by inserting an additional cross-attention layer between the feed-forward networks of each block of the Annotation Encoder. The output embedding is used as a multimodal representation of the sequence-annotation pair. Annotation Decoder adopts the same structure as Annotation Encoder. The input annotations (all masked to zero vectors) are combined with the sequence information injected through the cross-attention layer to predict the correct annotation list.

**Sequence-Annotation Matching (SAM) Filter:** This module consists of a simple linear layer that can identify whether the input sequence and annotation match by processing the fused features of the sequence-annotation pair. The output of the SAM filter $[P_{unmatch}, P_{match}]$ is a two-dimensional vector where the two dimensions represent the probability of a match and the probability of a mismatch of the input sequence and annotation pair, respectively.

## 3.2 UPSTREAM TASKS AND OBJECTIVES

**Overview:** We jointly optimize three objectives during training, one understanding-based objective and two generation-based objectives, and compute three losses to activate different modules, as shown below.

**Masked Language Modeling (MLM)** activates the Sequence Encoder. It aims to predict the identity of amino acids that have been randomly masked out of protein sequences:

$$\mathcal{L}_{\text{MLM}} = -\sum_{i \in M} \log p(x_i | x_{\backslash M}), \tag{1}$$

where for a randomly generated mask $M$ that includes 15% of positions $i$ in the sequence $x$, the model is tasked with predicting the identity of the amino acids $x_i$ in the mask from the surrounding context $x_{\backslash M}$, excluding the masked positions. This masked language modeling objective (Devlin et al., 2018) causes the model to learn dependencies between the amino acids. Although the training objective itself is simple and unsupervised, solving it over millions of evolutionarily diverse protein sequences requires the model to internalize sequence patterns across evolution.

**Sequence-Annotation Matching (SAM)** activates the Annotation Encoder. It aims to predict whether a pair of sequence and annotation is positive (matched) or negative (unmatched). Let $S$ denote input sequence and $A$ denote input annotation. We use Annotation Encoder's output embedding as the joint representation of the sequence-annotation pair, and append the SAM Filter to predict a two-class probability $p^{sam}$. The SAM loss is defined as the cross-entropy $H$ between $p$ and $y$:

$$\mathcal{L}_{\text{SAM}} = \mathbb{E}_{(S,A)\sim D} H(y^{\text{sam}}, p^{\text{sam}}(S, A)), \tag{2}$$

where $y^{sam}$ is a 2-dimensional one-hot vector representing the ground-truth label. We follow the strategy proposed in ALBEF (Li et al., 2021) to sample hard negatives for the SAM task with zero computational overhead. For each sequence in a mini-batch, we sample one negative annotation embedding from the same batch. Likewise, we also sample one hard negative sequence for each annotation. This results in the quantity of negative pairs being twice that of positive pairs for each mini-batch. Consequently, in practical training, we employ focal loss (Lin et al., 2017) in lieu of cross-entropy loss to mitigate the adverse effects arising from the imbalance in sample quantities.

**Annotation Prediction (AP)** activates the Annotation Decoder. This loss minimized by Annotation Decoder during training is a sum of the categorical cross-entropy over the protein sequences and the binary cross-entropy over the annotations, namely

$$\mathcal{L}_{\mathcal{AP}} = -\sum_{j\in N}(y_j^A \log(p_j^A) + (1 - y_j^A)\log(1 - p_j^A)), \tag{3}$$

where $N$ denotes the dictionary size of annotation list, $y_j^A \in \{0, 1\}$ is the true label for annotation $j$, and $p_j^A \in [0, 1]$ is the predicted probability that the protein has annotation $j$.

The **overall training objective** of ProtAC is:

$$\mathcal{L} = \min_\theta(\mathcal{L}_{\text{MLM}} + \mathcal{L}_{\text{SAM}} + \mathcal{L}_{\text{AP}}), \tag{4}$$

where $\theta$ denotes all trainable parameters including those of the three major modules and all projection heads. We minimize the loss functions of all upstream tasks simultaneously during training.

### 3.3 DATA CLEANING WORKFLOW

**Overview:** We propose a multi-round data cleaning strategy with three stages in each round, namely pretrain, finetune and caption. Our core aim is to facilitate a reciprocal enhancement of model performance and dataset quality through a cyclical process, wherein the initialized model goes through the pretraining stage and the finetuning stage, producing a pretrained model and a finetuned model, respectively. Both models are subject to evaluation covering upstream and downstream tasks. The pretrained model will replace the initialized model in subsequent cycles, while the finetuned model will enter the caption stage to perform data cleaning on the noisy dataset, thereby producing a cleaned dataset that will replace the noisy dataset in the next cycle (data cleaning workflow is shown in Fig.3).

**Stage Pretrain:** Our models will load last-round pretrained weights except in the first round where methods for initializing models vary across different versions. Pretraining dataset is the combination of Uniref50 (noisy dataset) and SwissProt-trainset (well-annotated dataset).

**Stage Finetune:** Our model inherits the weights from Stage Pretrain and is finetuned for 10 epochs on SwissProt-trainset, which enables the model to have a high ability to distinguish between fake and real protein annotations, so the model utilizes the knowledge gained during finetuning and performs well in Stage Caption. The pretrained or finetuned models are evaluated by downstream tasks.

**Stage Caption:** We use $[P_{unmatch}^{ori}, P_{match}^{ori}]$ and $[P_{unmatch}^{pred}, P_{match}^{pred}]$ to represent the SAM Filter output of original and predicted sequence-annotation pair. The SAM Filter of our finetuned model determines whether the original annotation or the model-predicted annotation is closer to the corresponding protein sequence through two key conditions:

1. The model predicts that the compatibility between the annotation and this specific protein sequence has been predicted to be positive, indicating a successful match, *i.e.* $P_{unmatch}^{pred} < P_{match}^{pred}$.

| Pretrained Model | Param. | GO-BP | | GO-MF | | GO-CC | | EC | |
|---|---|---|---|---|---|---|---|---|---|
| | | AUPR | $F_{max}$ | AUPR | $F_{max}$ | AUPR | $F_{max}$ | AUPR | $F_{max}$ |
| Param. >100M | | | | | | | | | |
| ProtBert (Elnaggar et al., 2021) | 420M | 0.188 | 0.279 | 0.464 | 0.456 | 0.234 | 0.408 | 0.859 | 0.838 |
| OntoProtein (Zhang et al., 2022) | 110M | 0.284 | 0.436 | 0.603 | 0.631 | 0.300 | 0.441 | 0.854 | 0.841 |
| ProtST-ESM-2 (Xu et al., 2023) | 782M | **0.342** | 0.482 | **0.647** | 0.668 | **0.364** | **0.487** | **0.898** | 0.878 |
| SaProt-650M (Su et al., 2023) | 650M | / | **0.486** | / | **0.682** | / | 0.479 | / | **0.882** |
| Param. <100M | | | | | | | | | |
| CNN (Shanehsazzadeh et al., 2020) | 38M | 0.165 | 0.244 | 0.380 | 0.354 | 0.261 | 0.387 | 0.540 | 0.545 |
| ResNet (Rao et al., 2019) | 6.5M | 0.166 | 0.280 | 0.281 | 0.267 | 0.266 | 0.403 | 0.137 | 0.187 |
| LSTM (Rao et al., 2019) | 28M | 0.130 | 0.248 | 0.100 | 0.166 | 0.150 | 0.320 | 0.032 | 0.082 |
| Transformer(Rao et al., 2019) | 38M | 0.135 | 0.257 | 0.172 | 0.240 | 0.170 | 0.380 | 0.187 | 0.219 |
| ESM2-8M (Rives et al., 2021) | 8M | 0.154 | 0.284 | 0.410 | 0.394 | 0.187 | 0.373 | 0.477 | 0.468 |
| ESM2-35M (Rives et al., 2021) | 35M | 0.212 | 0.340 | 0.501 | 0.489 | 0.248 | 0.417 | 0.562 | 0.571 |
| **ProtAC-PB** | 3M[*] | 0.139 | 0.221 | 0.350 | 0.327 | 0.180 | 0.254 | 0.410 | 0.424 |
| **ProtAC-ESM2-8M** | 8M[*] | 0.239 | 0.354 | 0.454 | 0.423 | 0.307 | 0.431 | 0.579 | 0.558 |
| **ProtAC-ESM2-35M** | 35M[*] | 0.268 | 0.379 | 0.577 | 0.603 | 0.321 | 0.461 | 0.615 | 0.619 |

Table 1: Downstream task performance. We use three color scales of blue to denote the first, second and third best performance in models < 100M and color with pink the overall best performance including models > 100M. *Abbr.*, PB: ProteinBERT; Param.: Parameter. [*] indicates the number of parameters we use for downstream tasks, see Tab. S1 for details of the full model. Performant PLM baselines are further introduced in the Appendix A.3 so as the discussion of performance difference.

2. The model-predicted annotation matches the protein sequence more closely than the original annotation, *i.e.* $P_{match}^{ori} < P_{match}^{pred}$ .

Only when both conditions are satisfied will original annotation be replaced by model-predicted annotation. The cleaned dataset will replace previous noisy dataset in the next round of Stage Pretrain.

# 4 EXPERIMENTS

## 4.1 EXPERIMENTAL SETUPS

**Datasets and Annotations:** We use **UniRef50** as of May 2018 for pre-training and captioning tasks, which contains 30.16 million protein sequences. **SwissProt**, updated to July 2023, contains 560,000 well-annotated sequences. We divide this database into two parts: 30,000 sequences are randomly selected for the test set, and the remaining approximately 530,000 sequences are used as the fine-tuning dataset. For the keyword prediction task, we further split the SwissProt test set in a 3:2 ratio, assigning 18,000 sequences to the training set and 12,000 sequences to the test set. The keywords associated with each sequence serve as labels to form the **Swiss-keyword** dataset. In the SwissProt caption task, we aggregate the newly updated sequences in SwissProt from 2023 to January 2024. We exclude sequences that overlap with sequences in the UniRef50 and SwissProt training datasets, resulting in a total of 458 sequences. This dataset is called the **Swiss-caption** dataset (see Appendix Tab.S3 for dataset details). We constructed annotation dictionaries of 7533 GO terms and 753 keywords, respectively (see Appendix B.1 for setup details).

**Model and Training Configurations:** We developed a ProteinBERT-based model and two ESM2-based models with different parameter versions: a small version using ESM2-8M and a basic version using ESM2-35M. Typically, we trained all models on eight A800 GPUs (time costs are shown in Fig. S6 and Tab. S8) with a training batch size of 256 for each model (equivalent to 32 proteins per GPU). We used the AdamW optimizer and an exponential learning rate scheduler, where the learning rate started at 1e-6, ramped up to 2e-5 during the first epoch, and then exponentially decreased back to 1e-6. Other settings are detailed in Appendix Tabs.S1 and S2.

**Downstream Benchmark Tasks:** To illustrate model capability in protein function annotation, we adopt two well-known benchmark tasks related to protein function, furthermore, we design a Keyword prediction task and a GO caption task using newly updated proteins in SwissProt:

- **Public Functional Annotation Tasks:** We adopted two established benchmarks introduced by DeepFRI (Gligorijević et al., 2021), specifically for Enzyme Commission (EC) number prediction and Gene Ontology (GO) term prediction. The GO benchmarks are divided into three different branches: molecular function (abbreviated as GO-MF), biological process (GO-BP), and cellular component (GO-CC).

- **Keyword Prediction Task:** Keywords (Magrane & Consortium, 2011) are another important form of protein function annotation. To evaluate the transfer learning ability of the model, we designed a classification task. The sequence encoder and annotation decoder of the pretrained model were frozen and the application layer was finetuned to evaluate keyword prediction.

- **Gene Ontology Caption Task:** To evaluate the model's ability to predict functions on never-seen sequences, we provide protein sequences and fully masked annotations from the Swiss-caption dataset as input to predict the corresponding GO annotations, which are then compared with newly curated GO terms from the SwissProt dataset.

## 4.2 EXPERIMENTAL RESULTS

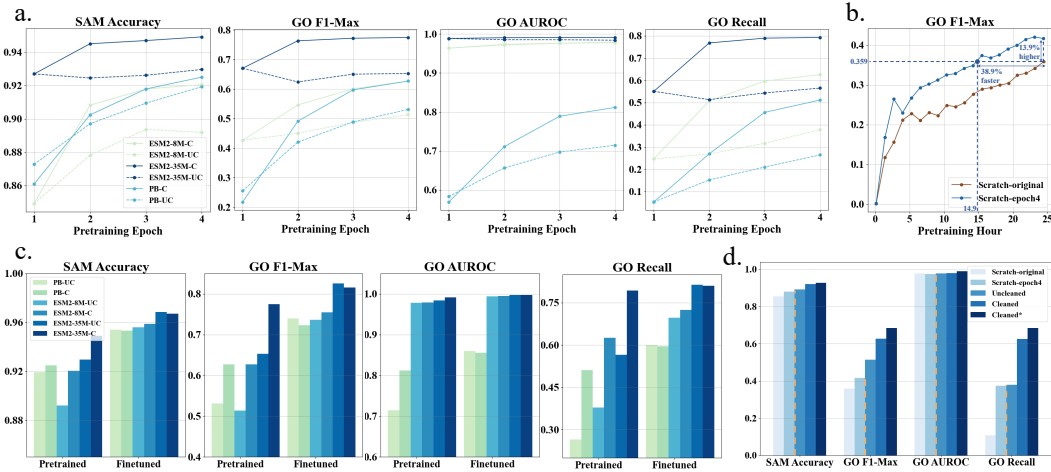

Figure 4: **(a)** Performance of different model versions during pretraining. The solid line represents the results of the model trained on the cleaned dataset, and the dashed line represents the results of the model trained on the original dataset. Each model was pretrained for four epochs and evaluated on the SwissProt test set after each epoch, using accuracy for SAM, F1-Max, AUROC, and recall for GO prediction. **(b)** Comparison of the improvement of pretraining on the original dataset and the cleaned dataset by GO prediction results. **(c)** Comparison between Finetuned and pretrained models. Cleaned: model trained on cleaned dataset; uncleaned: model trained on original dataset. The final results are presented after completing four rounds, where each round consists of pretraining the model for one epoch, followed by a finetuning phase of ten epochs. **(d)** Comparison of different pretraining strategies using ProtAC-ESM2-8M. Cleaned*: after the fourth epoch, the cleaned version is pretrained on the cleaned UniRef50 for one more full epoch. The dashed line demarcates two separate comparative analyses. *Abbr.*, C: Cleaned; UC: Uncleaned.

**Our model shows SOTA performance in protein function prediction tasks,** surpassing models with less than 100 million parameters and remaining competitive with larger PLMs. Tab.1 shows that our ProtAC-ESM2-35M ranks in the top three in four downstream tasks, achieves the highest score in GO-CC, and closely follows OntoProtein, a ProtBERT-based model with more than 400

million parameters, in GO-BP and GO-MF. In addition, our two ESM2-based models outperform their corresponding pretrained backbones in all tasks, and ProtAC-ESM2-8M not only outperforms ProtAC-PB, a model with similar number of parameters, but also surpasses ESM2-35M in three of the four tasks.

**Models with pretrained weights exhibit superior performance compared to those trained from scratch.** Fig.4a demonstrates that at an equivalent parameter count, the pretrained ProtAC-ESM2-8M model matches the SAM capability of the from-scratch ProtAC-PB model but significantly excels in GO prediction, indicating enhanced data cleaning efficiency through pretraining. Furthermore, as illustrated in Tab.2, comparisons between the 8M-cleaned and PB-cleaned versions reveal that, pretrained models consistently outperform their from-scratch counterparts.

**Larger models exhibit superior performance.** Fig.4a reveals that, among pretrained models, ProtAC-ESM2-35M outperforms ProtAC-ESM2-8M in both SAM and GO prediction, suggesting that increased model size enhances data cleaning efficacy. Additionally, Tab.2 shows that the 35M-cleaned model achieves the best outcomes across all four models, further supporting the conclusion that larger parameter sizes result in improved model performance.

**Our curated dataset demonstrates efficacy.** We applied Kaiming initialization (He et al., 2015) to the 8M-version model, followed by one epoch of pretraining using both the original dataset (Scratch-original) and a dataset refined through four cleaning cycles (Scratch-epoch4). Validation is conducted on the SwissProt test set during training (See Fig.4b). Our findings indicate that the maximum $F_{max}$ for Scratch-epoch4 surpassed that of Scratch-original by 13.9%, and notably, Scratch-epoch4 achieved this benchmark $F_{max}$ in 38.9% less training time. This evidence underscores the significant enhancement in the model's protein function prediction capabilities attributable to our meticulously cleaned dataset.

| Finetuned model | KW | |
|---|---|---|
| | AUROC | $F_{max}$ |
| ProtAC-ESM2-8M-cleaned | 0.8509 | 0.6474 |
| ProtAC-ESM2-8M-uncleaned | 0.8476 | 0.6424 |
| ProtAC-ESM2-35M-cleaned | **0.8602** | **0.6802** |
| ProtAC-ESM2-35M-uncleaned | 0.8533 | 0.6575 |
| ProtAC-ProteinBERT-cleaned | 0.7571 | 0.4871 |

Table 2: Keyword prediction results of ProtAC. In the main text, the four models listed in the table from top to bottom are referred to by the following abbreviated names: 8M-cleaned, 8M-uncleaned, 35M-cleaned, 35M-uncleaned, and PB-cleaned.

**Finetuned models surpass their pretrained counterparts in performance.** Fig.4c illustrates that finetuned models outshine pretrained ones across all four metrics. For the 8M-version models, those finetuned on cleaned datasets exhibit superior performance to their uncleaned counterparts; conversely, for PB-version models, those finetuned on original datasets fare better. The results indicate substantial enhancements in SAM and GO prediction for models trained on original datasets following finetuning. Nevertheless, the performance gains from finetuning are more pronounced for models with fewer parameters than for those with a larger parameter count.

**Our cleaning strategy yields positive results.** Fig.4a demonstrates that pretraining models on cleaned datasets enhances SAM and GO prediction capabilities compared to models pretrained on noisy datasets, indicating the effectiveness of our data cleaning approach for protein function prediction. Tab.2 further supports this observation by showing that models trained on cleaned datasets outperform those trained on original datasets of the same architecture. Additionally, our training approach utilizing different datasets is highlighted in Fig.4d, where Scratch-epoch4 outperforms Scratch-original in three out of four metrics, showcasing the continuous improvement in dataset quality facilitated by our cleaning strategy. Notably, the performance of the Cleaned* model excels in all four metrics, indicating that extended pretraining enhances the outcomes of our cleaning strategy (more comparison is shown in Appendix B.2).

**Data caption results show significant improvement.** Fig.5 shows that after training, our model's prediction performance in Swiss-caption has been significantly improved, and is far better than the original functional annotations of the dataset. Finetuned models generally outshine pretrained mod-

els in GO caption performance. Larger parameter sizes contribute to better performance, pretrained models outperform scratch-trained models, and the ProtAC-ESM2-35M-finetuned model stands out as the top performer among all versions. The quantities of sequences in UniRef50 cleaned by each model in every round are also delineated in Appendix Fig.S5.

### 4.3 BIOLOGICAL ANALYSIS

**GO annotation comparison for same protein**
To further validate the biological significance of the data cleaning results for ProtAC, we conduct a manual verification of the GO annotations before and after cleaning on a sampled set of protein data (see Appendix B.3.5 for detailed process). The newly added GO terms for the selected five clusters (Tab. S5; Tab. S9; Fig. 6a) are supported by evidence. For example, UniRef50_A0A1I4VGP3 (Fig. 6a, top) contains a member sequence, A0A1I4VGP3, which originally lacked corresponding GO annotations. After data cleaning with ProtAC, GO:0005886 is added in the first three rounds, and GO:0055085 is added in the fourth round. Upon review, the current GO annotation for this cluster in UniProt is GO:0016020. Both GO:0005886 and GO:0055085 are child terms of GO:0016020. Further investigation into the family and domain information for A0A1I4VGP3 reveals that it matches the IPR002549 family in the InterPro database. This family, known as the Transmembrane protein TqsA-like family, regulates quorum-sensing signal transmission by either enhancing the secretion of autoinducer-2 (AI-2) or inhibiting its uptake. This information suggests

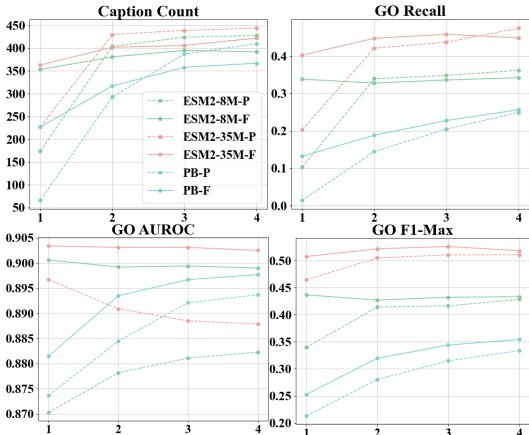

Figure 5: Protein function caption results across three model versions. The function of 418 sequences out of 458 has been captioned in newly updated SwissProt dataset. Notably, original GO yields an F1-Max of 0.0695, an AUROC of 0.1099, and a recall of 0.0344. *Abbr.*, P: Pretrained; F: Finetuned.

the potential inclusion of GO:0005886 and GO:0055085, indicating that ProtAC may enhance the granularity of protein annotations. Similarly, UniRef50_A0A1I1LTJ9 (Fig. 6a, bottom) contains two member sequences, both of which are annotated with GO:0016020 in the latest version of UniProt. Following ProtAC curation, GO:0016020 is added in the first round; in the second and third rounds, GO:0009881 and GO:0006355 are added; in the fourth round, GO:0006355 is removed, and GO:0030435 and GO:0046872 are added. Upon analysis, GO:0009881 and GO:0046872 are identified as co-occurring terms with GO:0016020, and these annotations are also supported by evidence from family and domain databases for the two member sequences of UniRef50_A0A1I1LTJ9.

**Protein comparison for same GO** We further analyze whether the protein sequences in clusters annotated with "transmembrane" related terms contained transmembrane regions. We first filter 152 GO annotations containing the term "transmembrane" from a total of 7,533 GO annotations in the GO dictionary. Subsequently, we randomly sample 20 clusters that contain neither of these 152 GO annotations before using ProtAC and contain some of these 152 GO annotations after the cleaning process. We use the Phobius Protein Functional Analysis tool [3] to predict transmembrane regions in all member sequences of the selected clusters. Among the 20 sampled clusters, 11 clusters are no longer present in UniRef50, and one cluster's sequences do not predict any transmembrane regions (Fig.6b). Transmembrane regions are predicted in the remaining 8 clusters. These results further underscore the biological relevance of ProtAC's performance in refining GO annotations (more biological analysis shown in Appendix B.3).

### 4.4 ABLATION STUDY

---

[3]https://www.ebi.ac.uk/jdispatcher/pfa/phobius

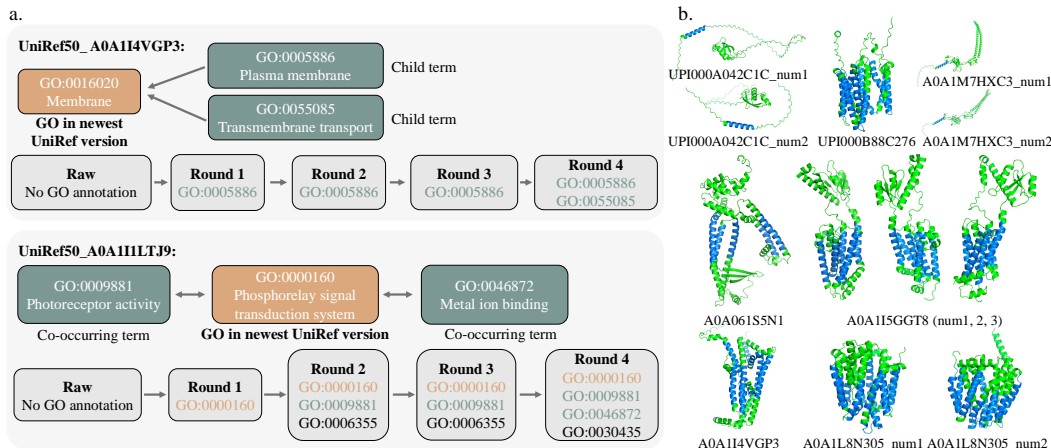

Figure 6: **(a)** Examples of GO variation analysis for the same protein in each cleaning round. We use two color to denote the GO annotations **in the newest UniRef** and **supported by family databases**. The child term would be a more specific term; the co-occurring terms would be co-annotated for the same protein or gene. **(b)** Predicted structures of transmembrane proteins. We use two different colors to distinguish between **transmembrane domains** and **other sequences**.

In the Experimental Results section, we have conducted comprehensive comparisons regarding the impact of the cleaned dataset, various cleaning strategies, and different model sizes on training effectiveness. Hence, in this section, we primarily investigate the influence of three losses on model performance. We pretrain ProtAC-PB for one epoch. Tab.3 shows that, eliminating $\mathcal{L}_{SAM}$ and $\mathcal{L}_{MLM}$, which are not directly related to function prediction, enhances the $F_{max}$ of pretraining GO and downstream Keyword prediction. However, removing any of the three losses significantly reduces the $F_{max}$ for GO captioning and downstream EC tasks. This underscores the critical importance of synthetic data filtration for model training (Shumailov et al., 2024).

| Model | Pre. GO | Cap. GO | EC | KW |
|---|---|---|---|---|
| Full loss | 0.2179 | 0.2132 | 0.4026 | 0.4282 |
| w/o $\mathcal{L}_{SAM}$ | 0.2488 | 0.0506 | 0.3348 | 0.4311 |
| w/o $\mathcal{L}_{MLM}$ | 0.3974 | 0.1114 | 0.3857 | 0.4607 |
| w/o $\mathcal{L}_{AP}$ | / | 0.0344 | 0.3477 | 0.3785 |

Table 3: Ablation study on three losses using pretrained models. We show the $F_{max}$ for four function prediction tasks. *Abbr.*, **Pre. GO**: GO prediction task in pretraining; **Cap. GO**: GO caption task. Notably, Pre. GO is dependent on $\mathcal{L}_{AP}$ for its functionality. Gray denotes the performance decay or vanishment compared with full loss.

## 5 CONCLUSION

We introduce ProtAC, a recursive cleaning framework that uses protein multimodal learning to optimize noisy annotations in large-scale protein datasets while enhancing the learning capabilities of PLMs. We develop a model with fewer than 100M parameters that achieves SOTA results on multiple function-related downstream tasks while also cleaning up a high-quality protein dataset.

However, limited by computational resources, our learning framework still has significant room for improvement. For instance, a protein's structure dictates its functionality, making structural information crucial for models to learn functional annotations accurately. Moreover, the vast research literature related to protein functions contains an abundance of extractable feature. Therefore, incorporating other modalities of information into our research is one of our future goals. In addition, this work has already demonstrated the immense potential of larger-scale PLMs, and we plan to explore the boundaries of protein function research further by introducing higher parameter-level PLMs in future studies. We also anticipate that the achievements of this work, including the curated dataset, can be applied to other tasks in protein representation learning, *e.g.* large-scale pretraining of protein models or *de novo* protein design.

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

# A MORE RELATED WORK

## A.1 LARGE-SCALE PROTEIN DATASETS

Large datasets can help to improve the scalability of the model and provide a more comprehensive representation of the underlying data distribution. They play a crucial role in protein generation such as the pretraining of sequence generation models (Alamdari et al., 2023; Zhang et al., 2023a; Gruver et al., 2024). Their comprehensive coverage of protein sequences from thousands of species enables the development of more robust and generalizable computational models. UniProt dataset (Magrane & Consortium, 2011) offers unparalleled representation of protein diversity sith over 200 million sequences from more than 20,000 species. This allows researchers to draw insights from a huge array of proteins. In contrast to specialized databases like Protein Data Bank(PDB) (Burley et al., 2019) and Gene Expression Omnibus (GEO) (Clough & Barrett, 2016) which focus on narrow data types, UniProt consolidates information from genomic, proteomic, and functional sources. This multi-modal view facilitates analysis of proteins from numerous angles. UniRef90, a protein sequence database that clusters sequences at 90 percents identity (Suzek et al., 2007), further enhances UniProt by reducing redundancy through sequence clustering. Similarly, UniRef50 is built by clustering UniRef90 seed sequences that have at least 50% sequence identity to and 80% overlap with the longest sequence in the cluster. The non-redundant sequences improve annotation quality and search efficiency. Regular updates also ensure researchers have access to the latest discoveries. By leveraging the scale and diversity of data in UniProt and UniRef, scientists can gain a deeper understanding of proteins and their many functions. These large-scale databases are foundational to modern bioinformatics.

## A.2 PROTEIN ANNOTATION DESCRIPTION

Comprehensively describing the diverse functions of proteins is critical for interpreting their roles in biological systems. While several annotation types exist, GO terms (Ashburner et al., 2000) and Keywords (Magrane & Consortium, 2011) are especially valuable. GO terms from the Gene Ontology allow consistent representation of molecular functions, biological processes, and cellular components across species. Their widespread use enables both granular annotation of individual proteins and higher-level pathway enrichment analysis. This dual utility makes GO terms a fundamental tool for functional genomics research (Huang et al., 2009). Keywords from UniProtKB similarly provide standardized vocabulary for protein functions. Manually curated for Swiss-Prot and automatically assigned for TrEMBL, Keywords capture multifaceted functional aspects in a structured ontology (Magrane & Consortium, 2011). The hierarchical organization into categories like molecular function and biological process aids literature indexing and database searching. By consolidating expert knowledge into controlled terminologies, GO terms and Keywords empower accurate computational analysis and biological interpretation. Their adoption throughout public bioinformatics databases highlights the indispensable role protein function annotation plays in translating sequence data into actionable knowledge.

## A.3 PERFORMANT BASELINE DISCUSSION

Here we introduce Performant baselines used in our work. ProtBert (Elnaggar et al., 2021), trained on massive protein databases, captured biophysical features and evolutionary information through self-supervised learning. ProteinBERT (Brandes et al., 2022) is a different model from ProtBert. We use it as one of our backbones. OntoProtein (Zhang et al., 2022) leverages knowledge graphs to integrate protein sequence and biomedical text information, achieving substantial improvements over ProtBert. ProtST (Xu et al., 2023) enhances protein language models by jointly learning from protein sequences and biomedical text as well. There baselines employ significantly larger parameters than ProtAC and captures connections between protein and functional annotations using efficient approaches like knowledge graphs and LLMs. SaProt (Su et al., 2023) is a large-scale protein language model that innovatively integrates both protein sequence and structural information through a novel structure-aware vocabulary system, achieving SOTA performance on protein prediction tasks. ProtT3 (Liu et al., 2024) is for Protein-to-Text Generation by incorporating a PLM as its protein understanding module and using a cross-modal projector to bridge the modality gap between proteins and text, since it is not applied on protein function prediction tasks, we do not consider it as a baseline in our work.

## B  ADDITIONAL EXPERIMENTAL RESULTS

### B.1  FUNCTIONAL ANNOTATION SETUPS

To construct an exhaustive Gene Ontology (GO) dictionary, we enumerated the number of occurrences of all GO terms in the UniRef and SwissProt datasets. We considered only those GO terms that appeared 100 times or more, resulting in a dictionary of 7533 terms (see Tab.S4 for composition analysis of the GO dictionary). The keyword dictionary included all keywords that appeared in the Swiss-keyword dataset, totaling 753 keywords. See Appendix A.2 for an overview of GO and keywords.

### B.2  ADDITIONAL EXPERIMENTS RELATED TO METHODOLOGY

#### B.2.1  FROM-SCRATCH TRAINING SETUPS

We developed two distinct parametric models based on ProteinBERT, labeled as "ProtAC-PB-small" and "ProtAC-PB-base". The "small" variant incorporates 6 layers and 4 attention heads, while the "base" model comprises 12 layers and 8 attention heads. These models underwent training on eight A800 GPUs, utilizing an AdamW optimizer in conjunction with a learning rate scheduler that includes a warm-up step followed by exponential decay. The initial learning rate was set to 1e-6, which was increased to 3e-4 during the warm-up phase, before being exponentially decreased back to 1e-6. The decay factor for the learning rate was maintained at 0.9, with the warm-up period lasting for 1 epoch. Moreover, the models were trained with a batch size of 128.

#### B.2.2  CLEANING STRATEGY WORKS FOR MODEL TRAINED FROM-SCRATCH

Fig. S1(a) illustrates the evolution of the base model's annotation prediction F1-score throughout the pretraining stage over three rounds. The graph demonstrates a progressive increase in the growth rate of the model's F1-score curve through successive cleanup cycles, coupled with a significant improvement in the peak value achieved. This pattern underscores the effectiveness of our data-cleaning strategy in enhancing the model's learning performance.

#### B.2.3  CONTINUOUS CLEANING STRATEGY

We explored two distinct data cleaning methodologies: the continue caption strategy and the not-continue caption strategy, obtaining valuable insights from both. Our validation approach comprised several steps: Initially, in Epoch 1, the small model was pretrained and fine-tuned using the uniref90 (original) dataset, followed by data cleaning to produce uniref90 (epoch1). Subsequently, in Epoch 2, the model was pretrained and fine-tuned on uniref90 (epoch1), and data cleaning was performed against both uniref90 (original) and uniref90 (epoch1) to create uniref90 (epoch2-nocontinue) and uniref90 (epoch2-continue). In Epoch 3, the model underwent pretraining and its training metrics were evaluated on uniref90 (epoch2-continue) and uniref90 (epoch2-nocontinue), respectively. Fig. S1(b) delineates the comparative analysis of the annotation prediction performance of the small model utilizing the no-continue caption and continue caption strategies. This figure illustrates two distinct curves that trace the maximum F1-score trajectories of the small model in Epoch 3, following pretraining on the uniref90 (epoch2-continue) and uniref90 (epoch2-nocontinue) datasets, correspondingly. The data clearly indicates that pretraining on the uniref90 (epoch2-continue) dataset results in a higher F1-score, thus underscoring the superior training effectiveness of the continue caption strategy. Based on these findings, the continue caption strategy was consistently employed for both training and data cleaning throughout our investigation.

#### B.2.4  IMPACT OF MODEL PARAMETERS ON CLEANING PROCESS TIME AND DATA CLEANING EFFICACY

Fig. S1(c) provides a comparative analysis of the temporal investment and testing performance across four iterative cleaning cycles for the small model and three cycles for the base model. The aggregate time spent per epoch by both models was meticulously recorded, and the refined models' annotation prediction proficiency was assessed using the SwissProt test set, employing the maximum F1-score metric. The outcomes of this assessment are depicted in a line graph. The analysis

indicates that the duration required for the small model to undergo four cleaning cycles is approximately two-thirds that of the base model's completion of three cycles. Furthermore, the small model exhibits superior F1-score and AUC values relative to the base model, suggesting that the small model achieves improved learning efficiency and outcomes over multiple iterations within a reduced timeframe.

### B.2.5 SEPARATE COMPARISONS OF ESM2 SERIES MODELS USING CLEANING STRATEGY WITH THE ORIGINAL PRETRAINED VERSION

As demonstrated in Tab. S6, which presents a comparison between two parameterized versions of ESM2 trained using our workflow, both versions exhibit significant enhancements in protein function prediction. This improvement underscores the efficacy of our training strategy in optimizing predictive performance.

### B.2.6 DETAILS ABOUT ADAPTIVE TRAINING

Inspired by works from Active learning (Sener & Savarese, 2017; Killamsetty et al., 2021; Mirzasoleiman et al., 2020), we designed adaptive training to enhance pretraining efficiency. We use $[P_{unmatch}^{ori}, P_{match}^{ori}]$ and $[P_{unmatch}^{pred}, P_{match}^{pred}]$ to represent the SAM Filter output of original and predicted sequence-annotation pair. We then use the condition $P_{match}^{ori} \geq P_{match}^{pred}$ to obtain a mask to filter samples where the model believes that the original annotations match the sequence better than the predicted annotations. These samples are intended to be further learned by the model, and we focus on updating only the loss they contribute in order to reduce the model's training time. This reduced the training time while ensuring or even improving the training effect (Tab. S7). We use pseudo code (see Algorithms 1) to explain the mechanism of Adaptive Training. We applied adaptive mask in the pre-training stage of each round (except the first round) and recorded the number of samples updated in each step (see Fig. S7). As cleaning rounds continue, the number of updated samples in the training step gradually decreases, and it can be seen that the noise level of the pretraining dataset is reduced.

## B.3 MORE BIOLOGICAL ANALYSIS

### B.3.1 COMPARISON OF VISUALIZATION RESULTS: ORIGINAL VS. CLEANED DATASET MODEL TRAINING

To verify that the cleaned dataset improves the performance of our model, we compared the clustering results of the model trained by the original Uniref vs. the model trained by the cleaned Uniref. We select protein sequences in *cellular component* GO domain in the Swissprot-test dataset. We then apply t-SNE to visualize the clustering of the seq embeddings from the model's sequence encoder. The clustering results (depicted in Fig. S2) demonstrate that embeddings derived from the model trained on the cleaned dataset exhibit significantly better coherence and separation. Taking three distinct subcellular compartments, namely cytosol (GO:0005829), extracellular region (GO:0005576), and nucleus (GO:0005634), which are spatially separated, as an example. We visualized the protein data that only contains annotations for one of these three compartments. It is evident that the embeddings obtained from models trained on the original data exhibit significant overlap, whereas the embeddings obtained from models trained on the cleaned data are distinctly separated from each other. This indicates that the cleaned dataset can enhance the model representative ability.

### B.3.2 COMPARATIVE VISUALIZATION OF ORIGINAL AND CLEANED SEQUENCE EMBEDDINGS

We conduct visualization analyses on the original dataset and the cleaned dataset, verifying the improvement of data quality. We first extract the sequences where the GO terms are revised after cleaning and then select a list of GO terms with a high number of occurrences. To ensure a fair comparison, we choose the model trained on SwissProt for embedding extraction. Then, we apply t-SNE to obtain the clustering outcomes of the original sequence embeddings vs. cleaned sequence embeddings. Fig. S3 reveals that the embeddings from the cleaned dataset result in markedly improved clustering, characterized by enhanced grouping and distinctiveness for sequences associated with the same GO terms. This outcome demonstrates the improved quality of the cleaned dataset.

### B.3.3 COMPARISION OF THE BIOPHYSICAL EMBEDDINGS OF AMINO ACIDS

The biophysical properties of amino acids, e.g., hydrophobicity, aromaticity, and charge, are widely recognized to profoundly impact the structural configurations of proteins. We visualize the biophysical properties of amino acids. Fig. S4 illustrates the comparative analysis of clustering outcomes between the model trained on the original dataset and one trained on the cleaned dataset. The findings indicate that models trained on the cleaned dataset show a slight improvement in clustering performance. In this regard, the distances between similar amino acids are more compact compared to before the cleaning process (e.g., hydrophobic (aromatic), polar neutral, positive amino acids), while there is a clear separation on the plane between hydrophobic amino acids and hydrophilic amino acids (positive and negative amino acids). This suggests that data cleaning significantly contributes to the model's ability to categorize the underlying biophysical characteristics of amino acids more effectively

### B.3.4 QUANTIFICATION OF NOISY LEVELS OF FUNCTIONAL ANNOTATIONS

In order to quantify the noise level of the cleaned dataset, we introduced Jaccard Similarity. We use it to compare the distribution similarity between the cleaned dataset and the ground truth dataset. The higher the similarity, the closer the cleaned dataset is to the true annotation, that is, the lower the noise level. We applied it on the Swiss-caption dataset. The Jaccard similarity coefficient measures the overlap between two sets by dividing the size of their intersection by the size of their union, shown as the following equation:

$$J(A, B) = \frac{|A \cap B|}{|A \cup B|} = \frac{\sum_{i=1}^{n} \min(A_i, B_i)}{\sum_{i=1}^{n} \max(A_i, B_i)} \tag{5}$$

Here $A$ denotes the cleaned dataset, $B$ denotes Swiss-caption, and the results are shown in Tab. S10. As the number of cleaning rounds continues, the similarity increases, which means that the noise level is decreasing.

### B.3.5 MANUAL CURATION OF GO ANNOTATIONS FOR IDENTICAL PROTEINS

The verification process involves the following steps: (1) random extracting 30 UniRef clusters with newly added GO terms after cleaning; (2) querying UniProt (as of 2024-07-30) for the existing GO annotations and associated family and domain information for sequences in these clusters; (3) verifying whether the new GO terms added by ProtAC correspond to updates in existing databases or to their ancestor or child terms, and assessing whether the family and domain information provides supportive evidence for the newly added GO terms. Among the randomly sampled 30 clusters, those that are either deprecated or have an excess of member sequences that can not be manually verified are excluded from the analysis.

| Model version | Seq encoder | Anno encoder | Layer | Head | Parameters |
|---|---|---|---|---|---|
| ProtAC-ProteinBERT | ProteinBERT local part | | 6 | 4 | 27M |
| ProtAC-ESM2-8M | ESM-8M | ProteinBERT global part | 6 | 4 | 29M |
| ProtAC-ESM2-35M | ESM-35M | | 12 | 8 | 78.56M |

Table S1: ProtAC model details

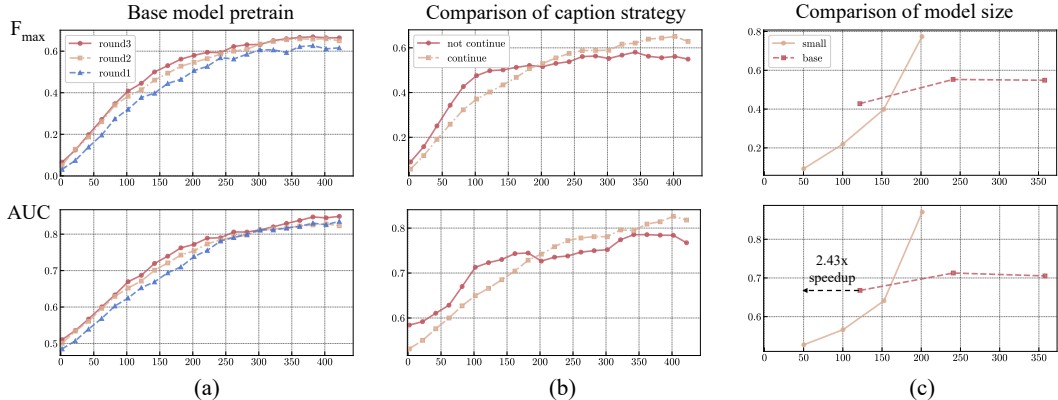

(a)    (b)    (c)

Figure S1: Visualisation of training results. The horizontal axis measures the training steps, where each step encompasses 1600 batches, and the vertical axis denotes the maximum F1-score achieved by the model in annotation prediction. (a) Annotation prediction curves of the base-model in the training phase of round1-3. (b) Comparison of the effects of different caption strategies on model training results. (c) Comparison of the learning effects of models at different scales.

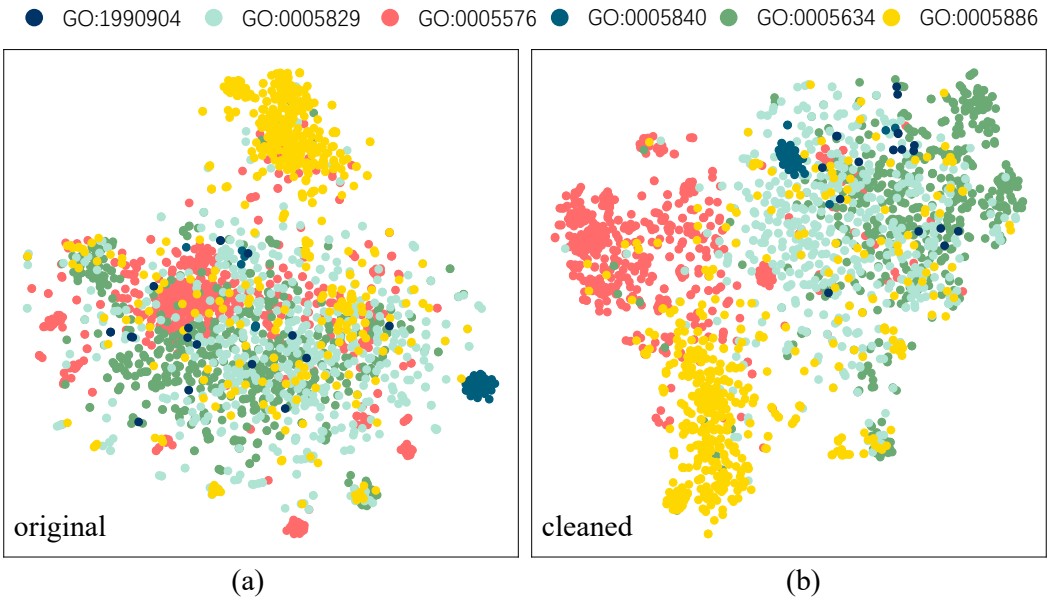

(a)    (b)

Figure S2: Visualization results by using the model trained on original (a) vs. cleaned dataset (b). The improved clustering outcome demonstrates that the cleaned dataset enhances the model representative ability.

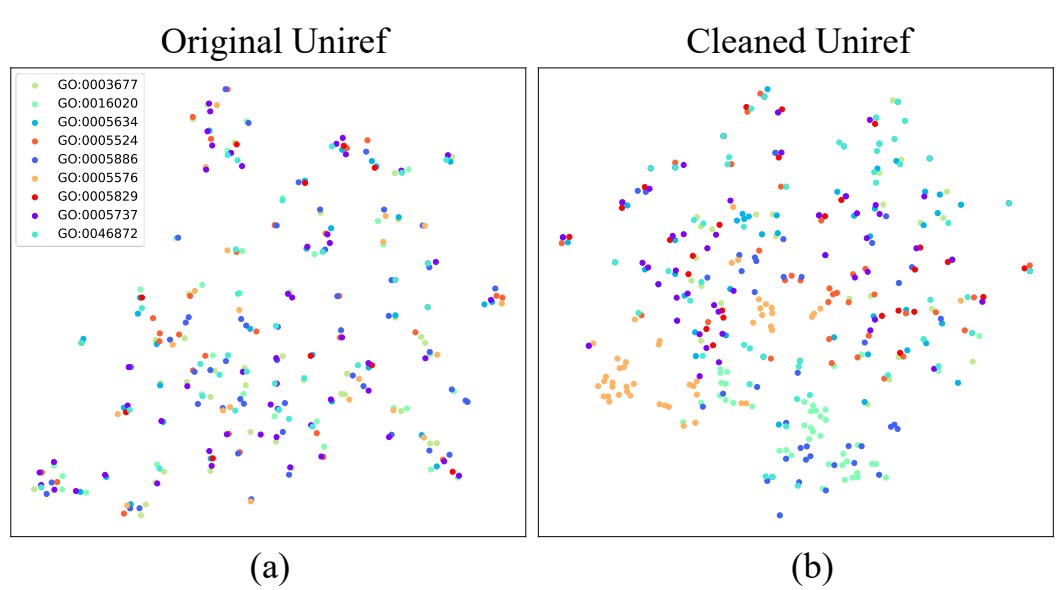

Figure S3: Visualization results of original vs. cleaned sequence embeddings. The cleaned dataset achieves better clustering results which validates the improved quality of the cleaned data.

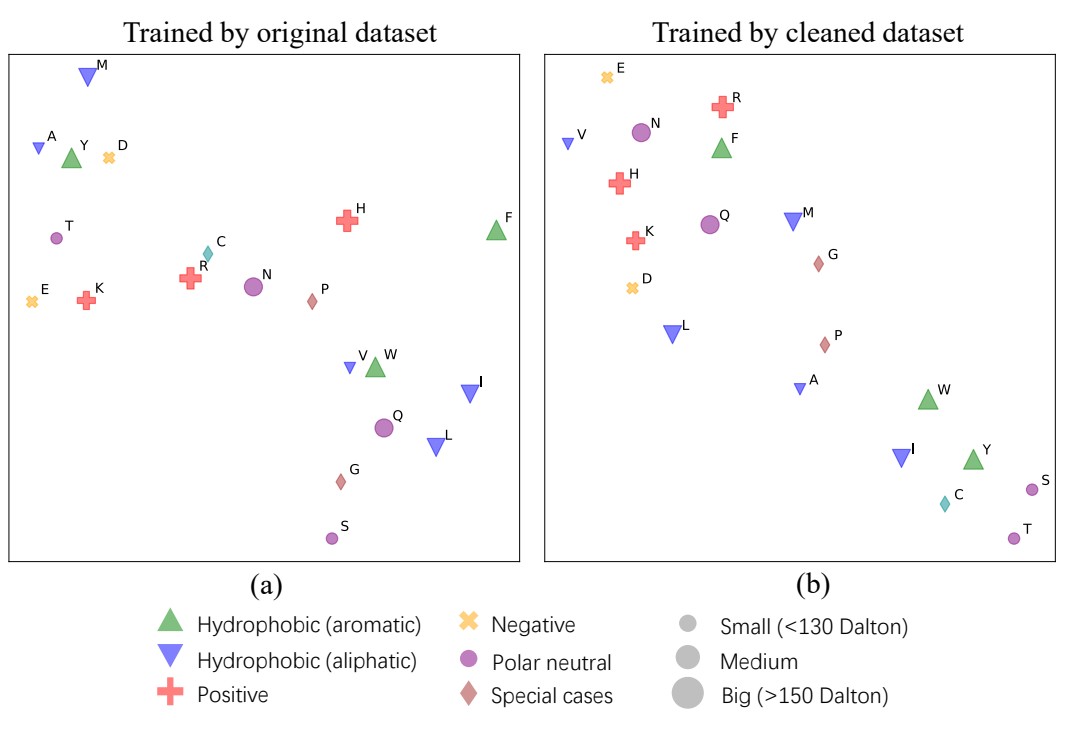

Figure S4: Biophysical embedding of amino acids.

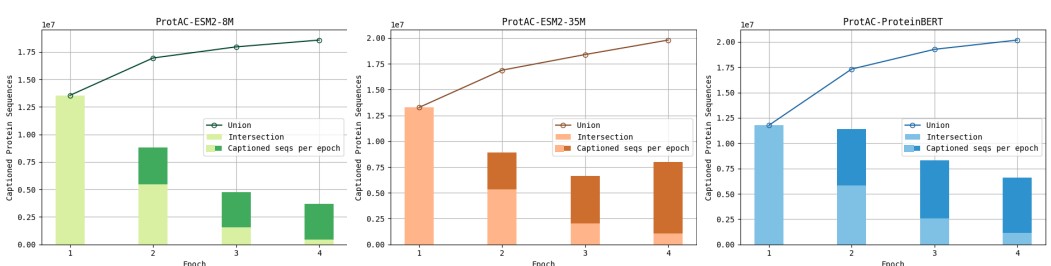

Figure S5: Captioned sequences count in UniRef50

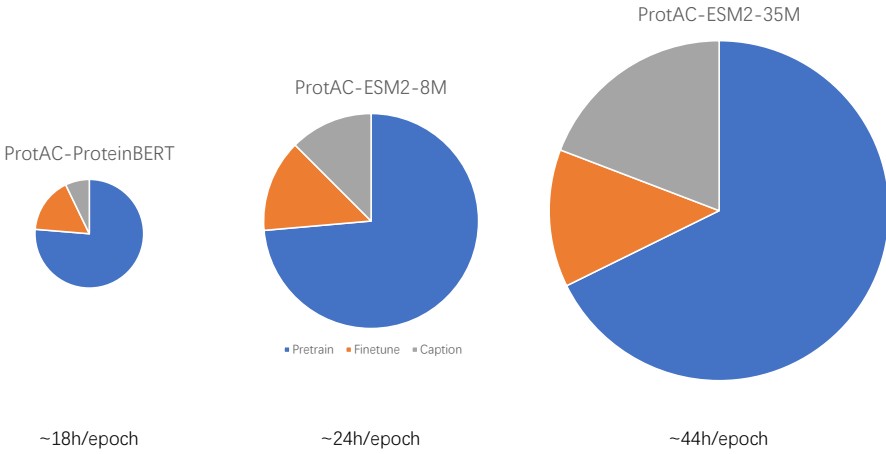

Figure S6: Time cost of each model version for one cleaning epoch/round

| Hyperparameter | Value |
|---|---|
| Input sequence length | 512 |
| Optimizer | AdamW |
| Scheduler | ExponentialLR |
| Pretraining epoch per round | 0.2 (ESM-8M) |
| | 1 (ProteinBERT&ESM-35M) |
| Finetuning epoch per round | 10 |
| Round | 4 |
| Pretraining dataset | UniRef50_2018_5 |
| Finetuning dataset | SwissProt_2023 |
| Init LR | 2e-5 |
| LR decay rate | 0.9 |
| Min LR | 1e-6 |
| Training batch size (8GPUs) | 512 |

Table S2: ProtAC training details

| UniRef50-2018 | SwissProt | | SwissProt(keyword) | | SwissProt(caption) |
|---|---|---|---|---|---|
| | trainset | testset | trainset | testset | |
| ~30.16M | ~530K | ~30K | 18K | 12K | 458 |

Table S3: Datasets details

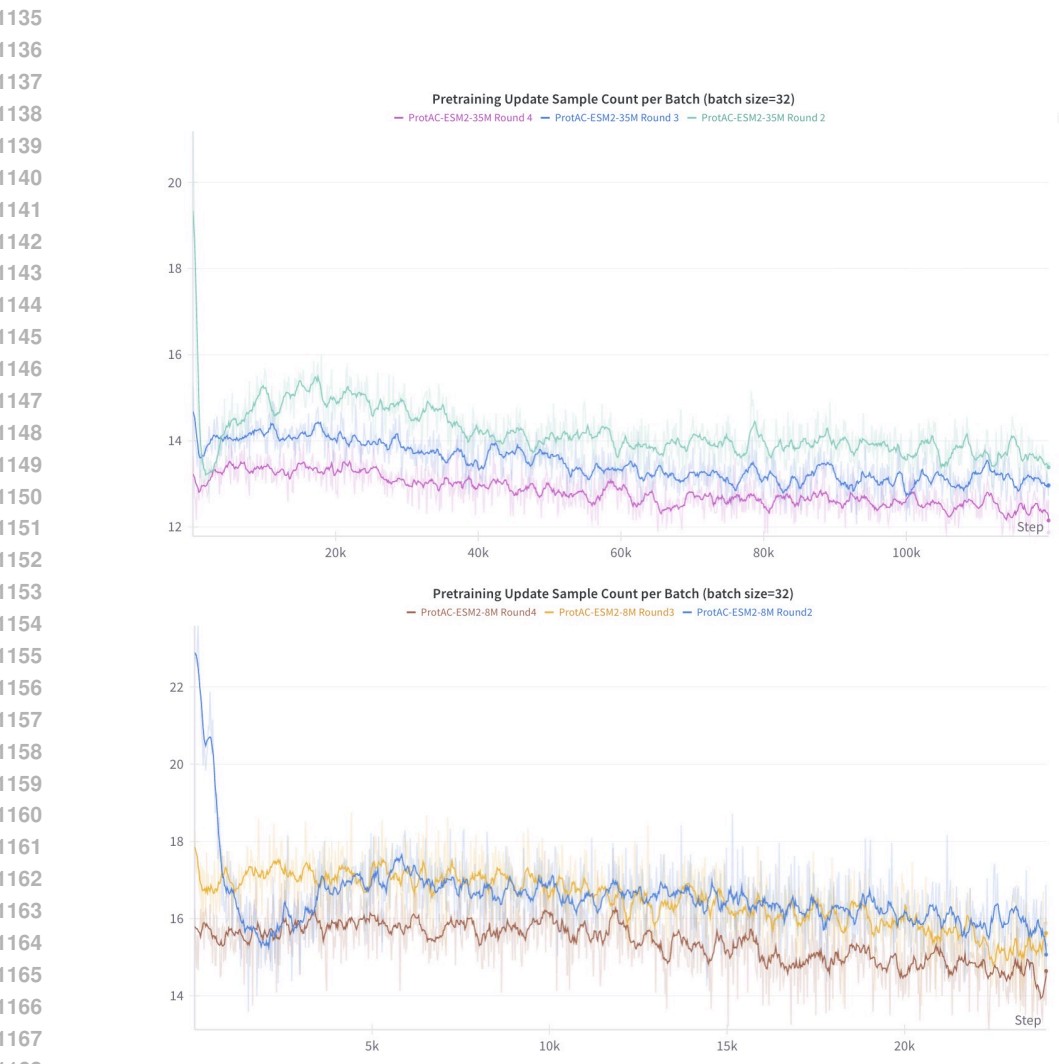

Figure S7: Pretraining samples of ProtAC-ESM2-8M and ProtAC-ESM2-35M updated in every step.

| GO Classification | Amount |
|---|---|
| CC (Cellular Component) | 962 |
| BP (Bioligical Process) | 3346 |
| MF (Molecular Function) | 3225 |
| Total | 7533 |

Table S4: GO dictionary details

| UniRef50 Cluster | Round 1 | Round 2 | Round 3 | Round 4 |
|---|---|---|---|---|
| A0A1I4VGP3 | GO:0005886 | GO:0005886 | GO:0005886 | GO:0005886; GO:0055085 |
| A0A1M6PSU4 | GO:0005886; GO:0009246; GO:0016874 | GO:0005886; GO:0009246; GO:0016874 | GO:0005886; GO:0009246; GO:0016874 | GO:0016020; GO:0016874; GO:0009103; GO:0005886; GO:0045227 |
| A0A1I1LTJ9 | GO:0000160 | GO:0009881; GO:0000160; GO:0006355 | GO:0009881; GO:0000160; GO:0006355 | GO:0009881; GO:0046872; GO:0030435; GO:0000160 |
| A0A1W9WW14 | - | - | GO:0016491; GO:0005783 | GO:0016491; GO:0005783 |
| A0A1M2W1M6 | - | - | GO:0016020; GO:0005783; GO:0009926; GO:0009734 | GO:0016020; GO:0005783; GO:0009926; GO:0009734 |

Table S5: GO comparison for same proteins. We use two color to denote the GO annotations **in the newest UniRef** and **supported by family databases**.

| Model | GO-BP | | GO-MF | | GO-CC | | EC | |
|---|---|---|---|---|---|---|---|---|
| | AUPR | $F_{max}$ | AUPR | $F_{max}$ | AUPR | $F_{max}$ | AUPR | $F_{max}$ |
| ESM2-8M | 0.154 | 0.284 | 0.410 | 0.394 | 0.187 | 0.373 | 0.477 | 0.468 |
| **ProtAC-ESM2-8M** | **0.239** ↑55.2% | **0.354** ↑24.6% | **0.454** ↑10.7% | **0.423** ↑7.4% | **0.307** ↑64.2% | **0.431** ↑15.5% | **0.579** ↑21.4% | **0.558** ↑19.2% |
| ESM2-35M | 0.212 | 0.340 | 0.501 | 0.489 | 0.248 | 0.417 | 0.562 | 0.571 |
| **ProtAC-ESM2-35M** | **0.268** ↑26.4% | **0.379** ↑11.5% | **0.577** ↑15.2% | **0.603** ↑23.3% | **0.321** ↑29.4% | **0.461** ↑10.6% | **0.615** ↑9.4% | **0.619** ↑8.4% |

Table S6: Separated comparisons between ESM2 and ProtAC-ESM2

| Model | Round | Adaptive Mask | Pretraning Time (Avg.)/h |
|---|---|---|---|
| **ProtAC-PB** | 1 | ✘ | 15.8 |
| | 2 ∼ 4 | ✔ | 13.8↓12.7% |
| **ProtAC-ESM2-8M** | 1 | ✘ | 17.8 |
| | 2 ∼ 4 | ✔ | 13.3↓25.3% |
| **ProtAC-ESM2-35M** | 1 | ✘ | 32.9 |
| | 2 ∼ 4 | ✔ | 30.2↓8.2% |

Table S7: Pretraining time comparison between Round 1 and rest Rounds

| Model | Pretraining/h | Finetuning/h | Caption/h |
|---|---|---|---|
| ProtAC-PB | 14.3 | 3.1 | 1.3 |
| ProtAC-ESM2-8M | 14.5 | 5.2 | 4.5 |
| ProtAC-ESM2-35M | 30.9 | 5.8 | 8.5 |

Table S8: ProtAC average time consumption for each stage

| UniRef50 Cluster | Round 1 | Round 2 | Round 3 | Round 4 |
|---|---|---|---|---|
| A0A2G5F261 | GO:0005886;
GO:0016020;
GO:0051119 | GO:0005886;
GO:0016020;
GO:0051119 | GO:0008643;
GO:0016020;
GO:0051119;
GO:0000139;
GO:0005886;
GO:0051260 | GO:0008643;
GO:0016020;
GO:0051119;
GO:0005886 |
| A0A1Y3BA38 | - | GO:0016020 | GO:0016020;
GO:0055085;
GO:0015031;
GO:0000329 | GO:0016020;
GO:0055085;
GO:0003333;
GO:0000329;
GO:0030435 |
| A0A2E6UZN8 | GO:0005886 | GO:0005886;
GO:0022857 | GO:0005886;
GO:0022857 | GO:0005886;
GO:0022857 |
| A0A1W9TJ44 | - | GO:0005886 | GO:0005886;
GO:0034755 | GO:0005886;
GO:0034755;
GO:0042597 |
| A0A1G6UTG6 | - | - | GO:0022857;
GO:0016020 | GO:0022857;
GO:0016020;
GO:0015562 |
| A0A1L7W0J8 | GO:0005886;
GO:0022857;
GO:0055085;
GO:0016020 | GO:0005886;
GO:0022857;
GO:0055085;
GO:0016020 | GO:0005886;
GO:0022857;
GO:0055085;
GO:0016020 | GO:0005886;
GO:0022857;
GO:0055085;
GO:0016020 |
| A0A1M6U3I0 | GO:0005886 | GO:0005886 | GO:0005886;
GO:0022857;
GO:0046677 | GO:0005886;
GO:0022857;
GO:0046677 |
| A0A1I3THQ2 | GO:0005886;
GO:0005267;
GO:0071805;
GO:0034765 | GO:0005886;
GO:0005267;
GO:0071805;
GO:0034765;
GO:0016020;
GO:0022841 | GO:0005886;
GO:0005267;
GO:0034765;
GO:0016020 | GO:0005244;
GO:0016020;
GO:0005516;
GO:0005267;
GO:0005886;
GO:0071805;
GO:0034765 |
| Q6ARZ3 | GO:0022857;
GO:0016020;
GO:0005886;
GO:0015562;
GO:0009279 | GO:0016020;
GO:0005886;
GO:0015562;
GO:0009279 | GO:0016020;
GO:0015562;
GO:0009279 | GO:0016020;
GO:0015562;
GO:0009279;
GO:0055085;
GO:0046677 |
| A0A1I0Q9V5 | GO:0046677;
GO:0022857;
GO:0055085;
GO:0005886;
GO:0140359 | GO:0046677;
GO:0022857;
GO:0055085;
GO:0005886;
GO:0140359 | GO:0046677;
GO:0022857;
GO:0055085;
GO:0005886;
GO:0140359 | GO:0046677;
GO:0022857;
GO:0055085;
GO:0005886;
GO:0140359 |
| A0A1M6TVP5 | GO:0005886;
GO:0022857;
GO:0015112;
GO:0042128 | GO:0005886;
GO:0022857;
GO:0015112;
GO:0042128 | GO:0005886;
GO:0008643;
GO:0015112;
GO:0042128 | GO:0005886;
GO:0008643;
GO:0015112;
GO:0042128 |

Table S9: The added comparison of GO annotations for identical proteins. We use two color to denote the GO annotations **in the newest UniRef** and **supported by family databases or belonging to the parent GO terms in the latest UniRef**.

| Model | Round | Jaccard Similarity |
|---|---|---|
| **ProtAC-PB** | 1 | 0.1011 |
| | 2 | 0.1475 |
| | 3 | 0.1594 |
| | 4 | 0.1744 |
| **ProtAC-ESM2-8M** | 1 | 0.1622 |
| | 2 | 0.2350 |
| | 3 | 0.2363 |
| | 4 | 0.2301 |
| **ProtAC-ESM2-35M** | 1 | 0.2425 |
| | 2 | 0.2862 |
| | 3 | 0.2821 |
| | 4 | 0.2820 |

Table S10: Noise level of Swiss-caption. The higher the value, the more similar the captioned annotation is to the ground truth distribution, that is, the lower the noise level.

---

**Algorithm 1** Adaptive Training

---

1: **Input:** $p^A, y^A, p^S$ {$p^A$: predicted annotation, $y^A$: original annotation, $p^S$: predicted sequence}

2: **Output:** $\mathcal{L}_{\mathcal{AP}}$
3: $[P^{pred}_{unmatch}, P^{pred}_{match}] \leftarrow SAM(AnnoEncoder(p^A, p^S))$
4: $[P^{ori}_{unmatch}, P^{ori}_{match}] \leftarrow SAM(AnnoEncoder(y^A, p^S))$
5: $Mask \leftarrow P^{ori}_{match} \geq P^{pred}_{match}$ {Create mask to filter samples}
6: **if** $Mask$ contains any True **then**
7: $\quad N_{update} \leftarrow \sum(Mask)$ {Count samples to be updated}
8: **end if**
9: $\mathcal{L}_{\mathcal{AP}} \leftarrow FocalLoss(p^A, y^A)$
10: $\mathcal{L}_{\mathcal{AP}} \leftarrow \mathcal{L}_{\mathcal{AP}}[Mask]$ {Apply mask to loss}
11: **return** $\mathcal{L}_{\mathcal{AP}}$

---

