# OpenReview forum: "Recursive Cleaning for Large-scale Protein Data via Multimodal Learning"
_ICLR.cc/2025/Conference — Submitted to ICLR 2025_

### Official Review · Reviewer_vPHH · 2024-11-02

**Soundness:** 2
**Presentation:** 2
**Contribution:** 2
**Rating:** 5
**Confidence:** 4

**Summary:**

This paper presents ProtAC, an innovative framework for recursive cleaning of large-scale protein datasets. ProtAC employs multimodal learning to enhance annotation accuracy by integrating sequence and functional data. The proposed Sequence-Annotation Matching (SAM) module refines annotations iteratively, using a pretrain-finetune-clean cycle to reduce noise and enhance dataset quality. This cycle leverages multimodal embeddings to identify and correct unreliable annotations in large datasets. The study claims state-of-the-art performance in protein function prediction across multiple benchmarks, achieving efficient and biologically meaningful annotations with models of relatively modest parameter sizes.

**Strengths:**

1. ProtAC introduces a recursive, multimodal cleaning framework for large-scale protein data, which combines sequence and functional annotation data in an iterative process. This recursive approach significantly improves data quality by progressively refining protein annotations, resulting in cleaner datasets and more accurate models.
2. By integrating sequence and functional data, ProtAC enhances annotation accuracy and functional prediction, addressing a common limitation in protein datasets. The multimodal Sequence-Annotation Matching (SAM) module offers a practical solution for aligning sequence data with functional information.
3. ProtAC achieves a state-of-the-art (SOTA) model with fewer than 100M parameters is achieved, outperforming competitors on multiple function-related downstream tasks. Additionally, it produces a cleaned version of the UniRef50 dataset, containing approximately 50 million accurately annotated protein sequences.
4. The study includes a detailed biological analysis of the cleaned dataset, showing that ProtAC’s modifications align well with established biological principles. This analysis underscores ProtAC’s potential for providing biologically meaningful insights, especially in the context of functional annotation.

**Weaknesses:**

While ProtAC introduces a promising strategy for improving dataset quality in protein bioinformatics, there are four weaknesses listed below that can be addressed to enhance its performance, generalizability, and applicability across diverse datasets and research objectives.

1. Despite emphasizing multimodal learning, ProtAC does not incorporate protein structure data, which is fundamental for accurate protein function prediction. By not incorporating structural data, the framework limits its understanding of functional nuances that are strongly influenced by three-dimensional protein conformations. This is critical for predicting protein functionality beyond simple sequence-based correlations, especially in cases where structural context dictates function (e.g., enzyme active sites). Integrating structural data would elevate the biological fidelity of predictions, improving ProtAC’s applicability in diverse biological and therapeutic contexts.
2. The pretrain-finetune-clean cycles require extensive computational resources and time, raising questions about the scalability of this approach to datasets beyond UniRef50，such as UniProt. Without scaling adaptations, ProtAC’s utility may be restricted to smaller databases, potentially underrepresenting protein diversity and thus limiting the reach of its cleaned annotations.
3. The approach heavily relies on a small, manually annotated dataset (SwissProt) for finetuning, which may limit generalization to diverse or rare protein functions. Relying on a manually curated SwissProt dataset introduces bias toward well-studied proteins, hindering generalization to underrepresented or novel sequences. This limitation affects ProtAC’s ability to accurately annotate functions in proteins from less-characterized species or environmental samples, reducing the utility of cleaned data in exploratory proteomics.
4. While competitive with larger models, ProtAC’s effectiveness may be further constrained by the limited model parameters (sub-100M), especially in highly complex functional predictions. The choice to use models under 100M parameters constrains ProtAC’s ability to capture complex biological interactions that may be better addressed by larger, more expressive models. Although it provides competitive performance on standard tasks, a parameter increase could unlock deeper representations that facilitate more accurate, granular annotations, particularly in multifunctional proteins.

**Questions:**

1. How do you plan to extend ProtAC to include structural data in future work?
2. Could you please discuss potential strategies for improving the efficiency of the proposed approach, and to provide estimates of the computational requirements for scaling to larger datasets like UniProt? How can the recursive cleaning process be optimized to balance effectiveness and computational cost?
3. Could you please explain how to address potential biases introduced by relying on SwissProt, and suggest ways to incorporate more diverse protein data in the finetuning process?
4. Could you please explain the rationale for choosing a parameter range of sub-100M? Whether you have explored larger models in future work?
5. In Table 1, OntoProtein shows substantial improvement over the original ProtBERT by leveraging knowledge graphs, effectively utilizing high-quality data. Could you elaborate on the unique advantages ProtAC offers in comparison to OntoProtein?
6. Table 2 indicates only a slight improvement in keyword prediction for the 8M-cleaned model compared to the 8M-uncleaned model. Could you provide results for the 35M-uncleaned model to offer a broader basis for comparison?
7. The ablation study results in Table 3 show inconsistent performance across the four tasks. Could you clarify whether these variations stem from the data-splitting method or the inherent differences between tasks?
8. In Figure S2, the clustering outcome for GO:0005576 on the cleaned dataset appears improved compared to the original, while the clustering for other terms shows limited improvement. Could you clarify why this effect is more evident for GO:0005576 and provide any additional insights or visualizations to highlight improvements for other terms?

**Details Of Ethics Concerns:**

No.

---

> ### Author Response · Authors · 2024-11-22
>
> ### Weakness 1 & Question 1 (structure modality)
>
> **W1:** *Despite emphasizing multimodal learning, ProtAC does not incorporate protein structure data, which is fundamental for accurate protein function prediction. By not incorporating structural data, the framework limits its understanding of functional nuances that are strongly influenced by three-dimensional protein conformations. This is critical for predicting protein functionality beyond simple sequence-based correlations, especially in cases where structural context dictates function (e.g., enzyme active sites). Integrating structural data would elevate the biological fidelity of predictions, improving ProtAC’s applicability in diverse biological and therapeutic contexts.*
>
> **Q1:** *How do you plan to extend ProtAC to include structural data in future work?*
>
> **A1:**
>
> Firstly, we strongly agree with reviewer vPH that structure ‘*is fundamental for accurate protein function prediction*’, but this does not mean that multimodal protein representation learning has to be adopted structure modality . In fact there are many great protein multimodal learning frameworks that also do not employ structural modality, such as OntoProtein [1], one of the baselines of this paper, which constructs a knowledge graph consisting of GO and its associated protein sequences and annotation texts for accurate function prediction; ProtST [2], one of the current sota models for protein function prediction, also does not directly use structures as inputs, but learns biomedical texts as features to fullfill contrastive loss with protein sequences. Therefore, we believe that the lack of structural modality cannot be a weakness of this paper.
>
> In line with this reviewer's view, we believe that the introduction of structural inputs is helpful for the optimisation of the data cleaning process as well as for the improvement of the predictive power of the model. We are glad that the reviewer raised the question **Q1** about how to introduce structural data in ProtAC, which is a very good question, as we mentioned in the conclusion section ‘incorporating other modalities of information into our research is one of our future goals’, so we are happy to share some premature ideas.
>
> For example, we can take the 3D structure data of proteins as an additional input, introduce a protein structure model such as GVP-GNN [3] as a structure encoder to learn and extract features, and then modify and update the three-modality contrastive loss [4] to achieve sequence-annotation-structure three-modality alignment. We can choose to pretrain the ProtAC of this paper first, and then freeze the weights of the sequence encoder and annotation encoder, and update the contrastive loss by backpropagation to train the structure encoder specifically. Ultimately, we design a filtering metric that combines the three modalities leading to more comprehensive data cleaning.
>
> Hopefully our immature ideas can provide some inspiration, and we also expect that this reviewer can provide more improvements based on our ideas to achieve a feasible solution！
>
> ***Reference:***
>
> [1]: Zhang N, Bi Z, Liang X, et al. Ontoprotein: Protein pretraining with gene ontology embedding[J]. arXiv preprint arXiv:2201.11147, 2022.
>
> [2]: Xu M, Yuan X, Miret S, et al. Protst: Multi-modality learning of protein sequences and biomedical texts[C]//International Conference on Machine Learning. PMLR, 2023: 38749-38767.
>
> [3]: Jing B, Eismann S, Suriana P, et al. Learning from protein structure with geometric vector perceptrons[J]. arXiv preprint arXiv:2009.01411, 2020.
>
> [4]: Xue L, Yu N, Zhang S, et al. Ulip-2: Towards scalable multimodal pre-training for 3d understanding[C]//Proceedings of the IEEE/CVF Conference on Computer Vision and Pattern Recognition. 2024: 27091-27101.

---

> ### Author Response · Authors · 2024-11-22
>
> ### Weakness 2 & Question 2 (resource demanding)
>
> **W2:** *The pretrain-finetune-clean cycles require extensive computational resources and time, raising questions about the scalability of this approach to datasets beyond UniRef50, such as UniProt. Without scaling adaptations, ProtAC’s utility may be restricted to smaller databases, potentially underrepresenting protein diversity and thus limiting the reach of its cleaned annotations.*
>
> **Q2:** *Could you please discuss potential strategies for improving the efficiency of the proposed approach, and to provide estimates of the computational requirements for scaling to larger datasets like UniProt? How can the recursive cleaning process be optimized to balance effectiveness and computational cost?*
>
> **A2:**
>
> Firstly we agree with reviewer vPH that efficiency is critical to applying our data cleaning solution to larger datasets, as this can help models understand protein diversity more fully. In fact, one of the main focuses of our work has been to improve the efficiency of data cleaning workflow, and we have achieved significant results. Model pre-training is the most time-consuming part of the entire data cleaning process. Fig. 4b shows that the 8M version trained on the cleaned dataset takes only **38.9% less time** than the model trained on the original to achieve the same accuracy and converge faster. Considering that the 8M version pre-training takes up nearly three-quarters of the total cleaning time, or about 18 hours (see **Fig. S6**), this **saves about 11 hours per round** compared to simply using the original dataset for training (assuming they can achieve the same prediction accuracy), which greatly shortens the time required for pre-training. We also filtered the data during the pre-training phase, selecting only data that met specific standards for training. This reduced the training time while ensuring or even improving the training effect.
>
> | **Model**       | Round | Adaptive Mask | **Pretraining Time (Avg.)/h** |
> | :-------------- | :---- | :------------ | :---------------------------- |
> | ProtAC-PB       | 1     | ✖             | 15.8                          |
> | ProtAC-PB       | 2~4   | ✔             | **13.8(-12.7%)**              |
> | ProtAC-ESM2-8M  | 1     | ✖             | 17.8                          |
> | ProtAC-ESM2-8M  | 2~4   | ✔             | **13.3(-25.3%)**              |
> | ProtAC-ESM2-35M | 1     | ✖             | 32.9                          |
> | ProtAC-ESM2-35M | 2~4   | ✔             | **30.2(-8.2%)**               |
>
> As can be seen from the above table (**Tab. S7**), the addition of adaptive mask effectively reduces the training time. We have added the detailed methodology and corresponding results for time-saving in the **Appendix B 2.6.**
>
> Of course, in future work we will continue to work on optimizing the details of the entire workflow and improving the efficiency and accuracy of data cleaning. According to the data on UniProt's official website, as of November 18, 2024, there are about 250M unreviewed protein sequences in UniProt's TrEMBL database. It is known that ProtAC-cleaned UniRef50-2018 has about 30M sequences. Assuming that the same four-round recursive cleaning strategy is adopted, theoretically, the time spent in the pre-training and cleaning stages will be about 8.3 times the current time. According to **Fig. S6 and Tab. S8**, after rough calculations, when using 8 A800 (80G) GPUs for training, **it will take about 530 hours to clean UniProt using the PB-version, about 652 hours using the ESM2-8M-version, and about 1331 hours using the ESM2-35M-version.**

---

> > ### Author Response · Authors · 2024-11-22
> >
> > ### Weakness 2 & Question 2 (resource demanding) continue
> >
> > **Q2:** *Could you please discuss potential strategies for improving the efficiency of the proposed approach, and to provide estimates of the computational requirements for scaling to larger datasets like UniProt? How can the recursive cleaning process be optimized to balance effectiveness and computational cost?*
> >
> > **A2:**
> >
> > We attach great importance to the balance between efficiency and computing resource consumption raised by the reviewer, because this is a very critical issue. Based on the above time estimate for cleaning UniProt, we can also see that our solution still has great prospects for optimization. Therefore, here we give three immature insights on the efficiency optimization of the strategy:
> >
> > 1. **Use a dataset with higher information density.** The main reason why the model pre-training phase takes a long time is that the pre-training dataset is too large (millions of sequences). If we can use less data (such as tens of thousands or even thousands of sequences) to achieve the same model level, the efficiency of data cleaning can be greatly improved. Therefore, it is very important to formulate quality metrics to deduplicate information in the dataset [1][2]. In fact, our large-scale protein dataset has a lot of information redundancy. If we filter out the most representative sequences, it is possible to achieve efficient pre-training with fewer resources.
> > 2. **Use lighter pretrained PLMs in the backbone.** Although the most advanced PLMs have billions of parameters, we believe that the current trend is for PLMs to scale down. Recently, many excellent lightweight PLMs have emerged, such as DistilProtBert [3], and Fastfold [4], which have demonstrated excellent capabilities in the field of protein learning. Therefore, we believe that choosing an excellent PLM with fewer parameters is a good choice to improve efficiency. Of course, this does not mean that we will give up those billion-level large models to pursue the ultimate performance.
> > 3. **Reduce recursive data cleaning rounds to only one round.** If we can optimize the current multi-round data cleaning strategy to only one round, the efficiency can be increased several times. In fact, continuous learning has been widely used in the field of large language models [5][6]. If we can apply this idea to the cleaning of protein data sets, and let PLM complete the identification and cleaning of noisy data while learning protein data, then the time required for data cleaning can be greatly shortened.
> >
> > We look forward to the reviewer's further feedback on the above discussion!
> >
> > ***Reference:***
> >
> > [1]: Nguyen T, Ilharco G, Wortsman M, et al. Quality not quantity: On the interaction between dataset design and robustness of clip[J]. Advances in Neural Information Processing Systems, 2022, 35: 21455-21469.
> >
> > [2]: Lee K, Ippolito D, Nystrom A, et al. Deduplicating training data makes language models better[J]. arXiv preprint arXiv:2107.06499, 2021.
> >
> > [3]: Geffen Y, Ofran Y, Unger R. DistilProtBert: a distilled protein language model used to distinguish between real proteins and their randomly shuffled counterparts[J]. Bioinformatics, 2022, 38(Supplement_2): ii95-ii98.
> >
> > [4]: Cheng S, Zhao X, Lu G, et al. Fastfold: Reducing alphafold training time from 11 days to 67 hours[J]. arXiv preprint arXiv:2203.00854, 2022.
> >
> > [5]: Wu T, Luo L, Li Y F, et al. Continual learning for large language models: A survey[J]. arXiv preprint arXiv:2402.01364, 2024.
> >
> > [6]: Bai Y, Jones A, Ndousse K, et al. Training a helpful and harmless assistant with reinforcement learning from human feedback[J]. arXiv preprint arXiv:2204.05862, 2022.

---

> > > ### Comment · Reviewer_vPHH · 2024-11-25
> > > **Follow-up concerns about A2**
> > >
> > > Thank you for your answers. However, I have some follow-up concerns:
> > > - **Regarding the 1st answer:**  How to boost the information density of the dataset? Could the author clarify what specific quality metrics or approaches can be used to identify and filter representative sequences, even though you have cited previous works [1],[2]? How would the reduction in dataset size (from millions to thousands of sequences) affect the diversity of data and generalizability of the pre-trained model in your work?
> > > - **Regarding the 2nd answer:** Could you provide evidence or benchmarks to compare lighter PLMs models to billion-parameter ones in your work? It is crucial to clarify that whether integrating lightweight models lead to performance degradation.
> > > - **Regarding the 3rd answer:** Could you provide any insights into the trade-offs between reducing cleaning rounds and maintaining data quality and data effectiveness in your tasks? How to ensure effectiveness when reducing the cleaning rounds to only one?

---

> > > > ### Author Response · Authors · 2024-11-26
> > > >
> > > > ### Follow-up answers to concerns about A2
> > > >
> > > > **Regarding the 1st answer**
> > > >
> > > > Thanks again to the reviewer for his reply and extended questions, which are of great help to further improve the quality of our article.
> > > >
> > > > First, **data deduplication** is one of the important means to improve the information density of the data set. As mentioned in [2], in the field of natural language processing, common data deduplication methods include: *EXACTSUBSTR* and *NEARDUP*, etc.
> > > >
> > > > However, these methods cannot be directly applied to protein learning. If two protein sequences have multiple fragments repeated, it does not mean that they can replace each other, because they may not be the same in some important fragments, resulting in completely different functions.
> > > >
> > > > Therefore, **for filtering**, we designed a method to screen representative sequences and apply it to the pre-training stage. In short, given an original sequence, we let the model predict the corresponding annotation and compare the matching probability of the predicted annotation and sequence $P_{match}^{pred}$ with the matching probability of the original annotation and sequence $P_{match}^{ori}$. Finally, all samples with $P_{match}^{ori} \geq P_{match}^{pred}$ are screened out. They represent seq-anno pairs that have not been fully learned by the model. The losses generated by them will be backpropagated (see **Appendix B.2.6** for details).
> > > >
> > > > In our work,we record the number of filtered training samples in each batch and use it as one of the indicators to **identify** the quality of the dataset (see **Fig. S7**). The fewer samples that need further training, the higher the quality of the dataset. On the other hand, if there is already a high-quality dataset, we can use it as a reference and use metrics such as ***Jaccard Index*** to calculate the similarity between sample sets to represent the quality of the cleaned dataset (we give the **relevant formula in Appendix B.3.4**). The Jaccard similarity coefficient measures the overlap between two sets by dividing the size of their intersection by the size of their union, shown as the following equation:
> > > > \begin{equation}
> > > >     J(A,B) = \frac{|A \cap B|}{|A \cup B|} = \frac{\sum_{i=1}^n \min(A_i, B_i)}{\sum_{i=1}^n \max(A_i, B_i)}
> > > > \end{equation}
> > > > Here $A$ denotes the cleaned dataset, $B$ denotes high-quality dataset (we supplemented an experiment measuring this metric at **Tab. S10** in the revised manuscript). The quality of dataset increases as similarity increases.
> > > >
> > > > Second, although there is data redundancy, we speculate that reducing the number of sequences from millions to thousands may reduce the diversity of the dataset and the generalization of the model to a certain extent, because they cannot cover the entire protein universe. But we must emphasize that this is a sacrifice of appropriate model generalization under the premise of striving for efficiency. [3] experimentally proved that when training multiple epochs, the utility of high-quality data is not great (because the model has completed learning). At this time, using lower-quality data (with lower utility at the beginning) is often more helpful than reusing high-quality data. The idea of alternating pretraining and finetuning in our workflow is also a potential solution to the quality-quantity tradeoff.
> > > >
> > > > Finally, we would like to emphasize that how to improve the information density of the dataset is very important for large model pre-training, but it is another very valuable issue that has not been explored in PLM pre-training. Since we have not explored this issue for a long time, we can only give some of our superficial views for discussion and inspiration, and we apologize for this.
> > > >
> > > > ***reference:***
> > > >
> > > > [2] Lee K, Ippolito D, Nystrom A, et al. Deduplicating training data makes language models better[J]. arXiv preprint arXiv:2107.06499, 2021.
> > > >
> > > > [3] Goyal S, Maini P, Lipton Z C, et al. Scaling Laws for Data Filtering--Data Curation cannot be Compute Agnostic[C]//Proceedings of the IEEE/CVF Conference on Computer Vision and Pattern Recognition. 2024: 22702-22711.

---

> > > > ### Author Response · Authors · 2024-11-26
> > > >
> > > > ### Follow-up answers to concerns about A2
> > > >
> > > > **Regarding the 2nd answer**
> > > >
> > > > Firstly, thanks to your reply! We would like to express our sincere apologies for not clearly articulating our perspective regarding "Using lighter pretrained PLMs in the backbone", which may have caused the reviewer to misunderstand our viewpoint. Please allow us to humbly clarify our viewpoint:
> > > >
> > > > Given our primary goal of computational efficiency and the practical constraints of limited computing resources, we thoughtfully adopted a compromise approach - selecting models with fewer parameters. While this choice may somewhat impact model performance (as larger models typically demonstrate superior task performance), it enabled us to conduct our research more efficiently within our resource constraints. To illustrate this point, we respectfully present comparative data on training time and downstream task performance (take GO-CC and EC as examples) between ProtAC-ESM2-8M and 35M shown in the table below.
> > > >
> > > > | Model           | Time cost on 8 A800 GPUs(4 rounds) $\downarrow$ | GO-CC Fmax $\uparrow$ | EC Fmax $\uparrow$ |
> > > > | --------------- | ----------------------------------------------- | --------------------- | ------------------ |
> > > > | ProtAC-ESM2-8M  | **96h**                                         | 0.431                 | 0.558              |
> > > > | ProtAC-ESM2-35M | 176h                                            | **0.461**             | **0.619**          |
> > > >
> > > > Due to current experimental limitations, **we have not yet been able to utilize billion-parameter PLMs in our work.** We humbly propose to expand the model scale in our future research endeavors.
> > > >
> > > > We sincerely apologize for any confusion our previous explanation may have caused, and we are deeply grateful for the reviewer's understanding and valuable feedback. We remain committed to improving our work based on these insights.

---

> > > > ### Author Response · Authors · 2024-11-26
> > > >
> > > > ### Follow-up answers to concerns about A2
> > > >
> > > > **Regarding the 3rd answer**
> > > >
> > > > We deeply appreciate the reviewer's insightful feedback. This line of inquiry indeed presents a promising research direction worthy of thorough investigation. We would like to propose the following methodological framework: leveraging state-of-the-art large-scale pretrained PLMs, such as ESM2-650M or even the 15B parameter model, as feature extractors. These sophisticated models have demonstrated remarkable capabilities in capturing both sequence patterns and functional characteristics with high fidelity.
> > > >
> > > > Our proposed approach involves implementing a Seq-Anno Matching module, followed by a two-phase training strategy. Initially, we would conduct fine-tuning on a carefully curated, high-quality dataset while maintaining the pretrained encoder weights frozen. Subsequently, we would integrate concurrent data cleaning procedures on an noisy dataset. This methodology aims to optimize both model performance and data quality simultaneously.
> > > >
> > > > This preliminary conceptual framework represents our current thinking on addressing the challenge, and we welcome further discourse on potential refinements and improvements.

---

> ### Author Response · Authors · 2024-11-22
>
> ### Weakness 3 & Question 3 (dataset bias)
>
> **W3:** *The approach heavily relies on a small, manually annotated dataset (SwissProt) for finetuning, which may limit generalization to diverse or rare protein functions. Relying on a manually curated SwissProt dataset introduces bias toward well-studied proteins, hindering generalization to underrepresented or novel sequences. This limitation affects ProtAC’s ability to accurately annotate functions in proteins from less-characterized species or environmental samples, reducing the utility of cleaned data in exploratory proteomics.*
>
> **Q3:** *Could you please explain how to address potential biases introduced by relying on SwissProt, and suggest ways to incorporate more diverse protein data in the finetuning process?*
>
> **A3:**
>
> We thank the reviewer for his concern about the potential bias caused by finetuning the model using SwissProt. However, we have a different view on the use of SwissProt. All sequence entries in the SwissProt database have been carefully verified by experienced molecular biologists and protein chemists using computer tools and by reviewing relevant literature [1]. They represent our most ground truth understanding of proteins. We need a dataset that represents the ground truth to fine-tune the model so that it has the ability to distinguish true from false functional annotations. SwissProt is the best candidate dataset to accomplish this mission.
>
> We also appreciate the reviewer's doubts about the diversity of SwissProt data, because it is limited by the number of included proteins (only about 570,000 sequences), and it is unlikely to represent the distribution of the entire protein universe, especially for some newly discovered proteins that have not been carefully studied. Therefore, in response to the reviewer's Q3 about addressing potential biases, our workflow adopts a strategy of alternating between large-scale dataset pre-training to provide generalization and small-scale dataset fine-tuning to provide authenticity; in addition, to enhance data diversity, we have proposed a possible solution: in the fine-tuning stage, add a dataset consisting of newly discovered proteins (such as those generated by evolutionary models [2][3][4] that are significantly different from known proteins in nature), and use the most advanced functional prediction models [5][6] to provide them with functional annotations, so as to reduce bias.
>
> We hope our answers can satisfy the reviewer's requirements. In addition, any new questions are welcome!
>
> ***Reference:***
>
> [1]: Boeckmann B, Bairoch A, Apweiler R, et al. The SWISS-PROT protein knowledgebase and its supplement TrEMBL in 2003[J]. Nucleic acids research, 2003, 31(1): 365-370.
>
> [2]: Rives A, Meier J, Sercu T, et al. Biological structure and function emerge from scaling unsupervised learning to 250 million protein sequences[J]. Proceedings of the National Academy of Sciences, 2021, 118(15): e2016239118.
>
> [3]: Lin Z, Akin H, Rao R, et al. Evolutionary-scale prediction of atomic-level protein structure with a language model[J]. Science, 2023, 379(6637): 1123-1130.
>
> [4]: Hayes T, Rao R, Akin H, et al. Simulating 500 million years of evolution with a language model[J]. bioRxiv, 2024: 2024.07. 01.600583.
>
> [5]: Wang S, You R, Liu Y, et al. NetGO 3.0: protein language model improves large-scale functional annotations[J]. Genomics, Proteomics & Bioinformatics, 2023, 21(2): 349-358.
>
> [6]: Zheng L, Shi S, Lu M, et al. AnnoPRO: a strategy for protein function annotation based on multi-scale protein representation and a hybrid deep learning of dual-path encoding[J]. Genome biology, 2024, 25(1): 41.

---

> ### Author Response · Authors · 2024-11-22
>
> ### Weakness 4 & Question 4 (model parameter)
>
> **W4:** *While competitive with larger models, ProtAC’s effectiveness may be further constrained by the limited model parameters (sub-100M), especially in highly complex functional predictions. The choice to use models under 100M parameters constrains ProtAC’s ability to capture complex biological interactions that may be better addressed by larger, more expressive models. Although it provides competitive performance on standard tasks, a parameter increase could unlock deeper representations that facilitate more accurate, granular annotations, particularly in multifunctional proteins.*
>
> **Q4:** *Could you please explain the rationale for choosing a parameter range of sub-100M? Whether you have explored larger models in future work?*
>
> **A4:**
>
> We are very pleased with this reviewer's suggestion to increase the number of model parameters. It is a very good proposal!
>
> Let's first answer the reason for choosing sub-100M: Most importantly, one of the core contributions of our work is to propose an efficient and feasible workflow for protein data cleaning. Pursuing a SOTA functional prediction model is not our main goal, so we don't need to over-stack the number of parameters; secondly, our computing resources are limited, and eight A800 GPUs are already the limit of our current available resources, so we must ensure that the size of the model and the training time are within an affordable range. Combining the above considerations and repeated trials and verifications, a model with about 100M parameters best meets the requirements of our task.
>
> Secondly, for larger model exploration, we used ESM2-150M as the backbone for a round of cleaning. The following is a comparison of the GO prediction results of the cleaned and uncleaned strategies. The better results achieved by the cleaned version prove the effectiveness of the cleaning strategy. However, since continuing the experiment will consume a lot of computing resources and time, this is not possible for us at this stage. However, the following results have proved the scalability of our framework. If you have any questions, please feel free to ask!
>
> | model (ProtAC-ESM2-150M)                 | SAM Accuracy  Pretrain | SAM Accuracy Finetune | GO F1 max Pretrain  | GO F1 max Finetune  | GO Recall Pretrain  | GO Recall Finetune  | GO AUROC Pretrain   | GO AUROC  Finetune  |
> | ---------------------------------------- | ---------------------- | --------------------- | ------------------- | ------------------- | ------------------- | ------------------- | ------------------- | ------------------- |
> | Round1                                   | 0.9272                 | 0.9637                | 0.6895              | 0.8173              | 0.4317              | 0.7951              | 0.9755              | 0.9886              |
> | Round2 Cleaned (compared with Round 1)   | 0.9285(+0.0013)        | **0.9677(+0.0040)**   | **0.7009(+0.0114)** | **0.8345(+0.0172)** | **0.6236(+0.1919)** | **0.8225(+0.0274)** | **0.9865(+0.0110)** | **0.9975(+0.0089)** |
> | Round2 UnCleaned (compared with Round 1) | **0.9298(+0.0026)**    | 0.9661(+0.0024)       | 0.6838(-0.0057)     | 0.8244(+0.0071)     | 0.6019(+0.1702)     | 0.8093(+0.0142)     | 0.9842(+0.0087)     | 0.9970(+0.0084)     |
>
> ###

---

> > ### Comment · Reviewer_vPHH · 2024-11-25
> > **Follow-up question to A4 (model parameter)**
> >
> > Thanks for your answer.
> > - As shown in the table above, although we see better results by using the cleaned version, the improvement is minor that hinder its effectiveness in practical use.

---

> > > ### Author Response · Authors · 2024-11-26
> > >
> > > ### **Follow-up answer to question concerning A4 (model parameter)**
> > >
> > > We are deeply grateful for the reviewer's thoughtful comments and would like to offer a detailed response to clarify our position.
> > >
> > > We find it imperative to highlight that the backbone model (pretrained ESM2-150M) referenced in Table A4 represents the culmination of extensive computational efforts and substantial GPU resources. As documented in [1], the training process was remarkably resource-intensive: "*We trained each model over 512 NVIDIA V100 GPUs. ESM2 700M took 8 days to train. The 3B parameter LM took 30 days. The 15B model took 60 days. All language models were trained for 500K updates, except the 15B language model which we stopped after 270K updates due to computational constraints.*"
> > >
> > > Given the comprehensive nature of this pretraining, achieving even marginal improvements upon these results presents a significant challenge. This observation substantiates our paper's assertion regarding the advantages of utilizing pretrained models.
> > >
> > > To further demonstrate our method's effectiveness on models trained from scratch, we conducted additional experiments by initializing ESM2-150M and training it with both the original and cleaned datasets. Given our relatively limited computational resources (approximately 1% of those available to the ESM team) and time constraints, **we pretrained for one tenth of a pretraining stage on UniRef50 dataset (approximately three million sequences)**. The resultant GO prediction outcomes are presented in the accompanying table, which demonstrates that the cleaned data substantially enhances the model's predictive capabilities.
> > >
> > > | ProtAC-ESM2-150M from scratch      | GO Fmax    | GO Recall  | GO AUROC   | SAM Accuracy |
> > > | ---------------------------------- | ---------- | ---------- | ---------- | ------------ |
> > > | pretrained on **Cleaned** dataset  | **0.2483** | **0.1193** | **0.9374** | **0.7349**   |
> > > | pretrained on **Original** dataset | 0.2130     | 0.0216     | 0.9089     | 0.6836       |
> > >
> > > We sincerely appreciate the reviewer's rigorous examination of our work and hope that these additional clarifications address the concerns raised. We remain open to further discussion and feedback to improve our research.
> > >
> > > ***Reference:***
> > >
> > > [1] Lin Z, Akin H, Rao R, et al. Evolutionary-scale prediction of atomic-level protein structure with a language model[J]. Science, 2023, 379(6637): 1123-1130.

---

> > > > ### Author Response · Authors · 2024-12-03
> > > >
> > > > ### Supplementary experimental results for question A4 (model parameter)
> > > >
> > > > Dear reviewer vPHH,
> > > >
> > > > We appreciate your valuable feedback regarding model parameter scaling. As suggested, we conducted additional experiments by training two models of 150M parameter version from scratch, comparing performance between the cleaned and original datasets. The training has progressed from 10% to 50% completion (approximately 15 million sequences), results are shown in the table below.
> > > >
> > > > | ProtAC-ESM2-150M from scratch     | GO Fmax $\uparrow$ |            | GO Recall $\uparrow$ |            | GO AUROC $\uparrow$ |            | SAM Accuracy $\uparrow$ |            |
> > > > | --------------------------------- | ------------------ | ---------- | -------------------- | ---------- | ------------------- | ---------- | ----------------------- | ---------- |
> > > > | **Training process in one epoch** | 10%                | **50%**    | 10%                  | **50%**    | 10%                 | **50%**    | 10%                     | **50%**    |
> > > > | w/ **cleaned** dataset            | 0.2483             | **0.3410** | 0.1193               | **0.2299** | 0.9374              | **0.9697** | 0.7349                  | **0.8489** |
> > > > | w/ **original** dataset           | 0.2130             | 0.3098     | 0.0216               | 0.0631     | 0.9089              | 0.9543     | 0.6836                  | 0.8227     |
> > > >
> > > > The comparative results demonstrate that across all four evaluation metrics, the model trained on the cleaned dataset consistently outperforms its counterpart trained on the original dataset. These findings not only validate the effectiveness of our data cleaning strategy but also, in conjunction with the results presented in **A4**, demonstrate the scalability of our framework to larger parameter models.
> > > >
> > > > While the experiments required nearly a week to complete due to computational constraints, we believe these new results substantively address your concerns about scalability and cleaning effectiveness. We sincerely thank you for your thorough review, which has helped us improve the quality of our work significantly.
> > > >
> > > > Given these additional experimental validations, we would be grateful if you could consider updating your rating score!

---

> ### Author Response · Authors · 2024-11-22
>
> ### Question 5 (model advantages)
>
> **Q5:** *In Table 1, OntoProtein shows substantial improvement over the original ProtBERT by leveraging knowledge graphs, effectively utilizing high-quality data. Could you elaborate on the unique advantages ProtAC offers in comparison to OntoProtein?*
>
> **A5:**
>
> We appreciate the comparison with OntoProtein. While OntoProtein indeed shows impressive results through knowledge graph integration, ProtAC offers several distinct advantages:
>
> - ProtAC introduces a novel recursive cleaning framework that actively improves data quality during training, rather than solely relying on existing knowledge graphs. This self-improving mechanism allows the model to learn from and refine noisy data.
> - Our Sequence-Annotation Matching (SAM) module provides explicit verification of sequence-function relationships, enabling more reliable function predictions compared to knowledge graph embedding alone.
> - ProtAC achieves competitive performance with a significantly smaller parameter count (<100M), making it more computationally efficient and accessible than larger models.

---

> ### Author Response · Authors · 2024-11-22
>
> ### Question 6 (experiment comparison)
>
> **Q6:** *Table 2 indicates only a slight improvement in keyword prediction for the 8M-cleaned model compared to the 8M-uncleaned model. Could you provide results for the 35M-uncleaned model to offer a broader basis for comparison?*
>
> **A6:**
>
> Thanks to this reviewer for his great suggestion on the completeness of the experimental comparison! We have added the keyword prediction results of the 35M-uncleaned model in Tab. 2. In addition, we have also added the GO prediction results of the 35M-uncleaned model in Fig. 4. The reviewer can check them together in the revised manuscript!

---

> ### Author Response · Authors · 2024-11-22
>
> ### Question 7 (ablation study)
>
> **Q7:** *The ablation study results in Table 3 show inconsistent performance across the four tasks. Could you clarify whether these variations stem from the data-splitting method or the inherent differences between tasks?*
>
> **A7:**
>
> We believe that inconsistent performance across the four tasks arises from inherent differences between tasks. First of all, AP loss is strongly related to annotation prediction, so its absence will lead to a serious decrease in the accuracy of the prediction task; secondly, MLM loss and SAM loss are not directly related to function prediction, and their absence will allow the model to focus on function prediction,  improving the accuracy of GO and KW prediction. However, the cleaning of functional annotations requires the model's understanding of the sequence and the model's ability to match sequence-annotation pairs. Therefore, the lack of MLM loss and SAM loss will lead to a significant reduction in the accuracy of GO caption.
>
> In order to eliminate the possibility that the inconsistency of the results is caused by the dataset division, we selected a task: keyword prediction, and re-divided the data set. We re-divided the Swiss-keyword data set into training and test sets in a ratio of 5:1, and re-trained the ablation models of ProtAC-PB. The prediction results show that the data set division is not the main reason affecting the experimental results, because the new results are in the same order (from best to worst performer) as before.
>
> | Task: KW prediction | Split        | full loss      | w/o SAM         | w/o MLM           | w/o AP         |
> | ------------------- | ------------ | -------------- | --------------- | ----------------- | -------------- |
> | F1-max              | 3:2 (origin) | 0.4282 (third) | 0.4311 (second) | **0.4607 (best)** | 0.3785 (worst) |
> | F1-max              | 5:1 (new)    | 0.4777 (third) | 0.5526 (second) | **0.6223 (best)** | 0.4731 (worst) |

---

> ### Author Response · Authors · 2024-11-22
>
> ### Question 8 (GO clustering)
>
> **Q8:** *In Figure S2, the clustering outcome for GO:0005576 on the cleaned dataset appears improved compared to the original, while the clustering for other terms shows limited improvement. Could you clarify why this effect is more evident for GO:0005576 and provide any additional insights or visualizations to highlight improvements for other terms?*
>
> **A8:**
>
> The protein functional annotations shown in **Fig. S2** are as follows: 1) GO:1990904: ribonucleoprotein complex; 2) GO:0005829: cytosol; 3) GO:0005576: extracellular region; 4) GO:0005840: ribosome; 5) GO:0005634: nucleus; 6) GO:0005886: plasma membrane. Among these, the cellular compartment represented by GO:0005576 (extracellular region) is spatially distinct from intracellular compartments (GO:1990904, GO:0005829, GO:0005840, GO:0005634), and proteins annotated with both GO:0005576 and other cellular localizations are less common. Therefore, a notable improvement in performance is observed for GO:0005576 in terms of data cleaning. Additionally, due to the spatial distinction of proteins annotated with GO:0005886 (plasma membrane), the visualization of this term also shows better results compared to other annotations. Furthermore, as shown in Figure S2, GO:0005829 and GO:0005634 are harder to distinguish in the dimensionality reduction visualization. However, many proteins annotated with GO:0005634 (nucleus) often have multiple localizations. These proteins may primarily function in the nucleus, but they can also play auxiliary roles in other cellular structures (such as the cytoplasm). Thus, many proteins annotated with GO:0005634 may also be annotated with other GO terms representing different cellular localizations.

---

> ### Author Response · Authors · 2024-11-27
> **Request for Follow-up on Rebuttal Response**
>
> Dear Reviewer vPHH,
>
> Thank you for your previous comments, to which we have provided detailed responses. If you find our responses satisfactory, we would greatly appreciate if this could be reflected in the revised scores.
>
> We noticed that **the deadline for submitting our revised PDF is approaching**, and we would be very grateful if you could share your thoughts on our responses at your earliest convenience. Your dedication to the review process is deeply appreciated.
>
> Thank you again for your time and effort.
>
> Best regards,
>
> Authors

---

### Official Review · Reviewer_S9Fj · 2024-11-02

**Soundness:** 3
**Presentation:** 3
**Contribution:** 2
**Rating:** 6
**Confidence:** 3

**Summary:**

This paper introduces a novel method called ProtAC, which applies a scalable automatic cleaning framework leverage both Protein sequence and functional information through multimodal learning to correct large protein datasets. Then, it conducts experiments on multiple function-related downstream tasks and publish a cleaned UniRef50 dataset containing ~50M proteins with well-annotated functions. Furthermore, it also presents experiment that demonstrate biological meaningful significance.

**Strengths:**

1. This methods propose the Sequence-Annotation Matching module and a cyclic process which consists of three stages：”train-finetune-clean” cycles. In this way, it may clean the large protein datasets and obtain an effective dataset, which provide a novel method for other researchers to clean the dataset they need.
2. In Table 1, some protein function prediction task performance shows that ProtAc+backbone surpassing models with less more than 100 million parameters.

**Weaknesses:**

Some technical details should be explained, please refer to Question part.

**Questions:**

1. In Table 1, the ProtAC-PB, does the pharse PB refer to ProtBert in your baseline？In my view, we find that the performance of ProtAC-PB is lower than ProtBert. Could you explain what caused this? And the baseline that you choose is not novel, you are supposed to compare with the latest baseline. Does your method have been tested on SaProt [1]? if not, you are supposed to test on SaProt as a way to strengthen the evaluation.
2. In Table 2, as you described in Section 4.2 “Furthermore, as illustrated in Tab 2 comparisons between the 8M-cleaned and PB-cleaned version reveal that, pretrained models consistently outperform their from-scratch counterparts”, whether different parameters size cause this phenomenon？, we find that lacking the ProtAC-ESM2-35M-cleaned models, why don’t you compare their performance？You could include results for the in ProtAC-ESM2-35M-cleaned models.
3. During the process of cleaning the large protein datasets, does this process cause additional noise to influence the quality of cleaned datasets. You are supposed to explain how to reduce additional noise, or quantify the quality/noise level of the cleaned datasets compared to the original.
4. In addition, authors should provide access to code and datasets so that other researchers can reproduce the results.

[1] Su J, Han C, Zhou Y, et al. SaProt: Protein Language Modeling with Structure-aware Vocabulary[C]//The Twelfth International Conference on Learning Representations.

---

> ### Author Response · Authors · 2024-11-22
>
> ### Question 1 (more baseline)
>
> **Q1:** *In Table 1, the ProtAC-PB, does the pharse PB refer to ProtBert in your baseline？In my view, we find that the performance of ProtAC-PB is lower than ProtBert. Could you explain what caused this? And the baseline that you choose is not novel, you are supposed to compare with the latest baseline. Does your method have been tested on SaProt [1]? if not, you are supposed to test on SaProt as a way to strengthen the evaluation.*
>
> **A1:**
>
> Thanks for the reviewer's question. We sincerely apologize for the unclear statement. PB and ProtBert are not the same model. PB stands for ProteinBERT [1] (we have stated this abbreviation in the Tab1 caption). It is a PLM with a similar number of parameters to ESM2-8M. We use it as one of our backbones and train it to compare the performance gap between the pre-trained model and the de novo training model. ProtBert [2] is a performant PLM with 420M parameters. We mainly use it as one of the most sota baselines for protein function prediction. Given the huge difference in parameter size between the two models, we believe the results are reasonable. We will distinguish them again in **Appendix A.3**. Thank you for your contribution to the clarity of our article!
>
> Thanks to the reviewer for the constructive suggestions on the completeness of the baseline, we have included the functional prediction task results of SaProt in **Tab. 1**.  SaProt is an excellent multimodal PLM. Unlike all other baselines we compared, it takes the three-dimensional structure of the protein as one of the inputs, which strongly proves the importance of structure for protein function learning. We have added a detailed introduction to all performant PLM baselines (including SaProt) in **Appendix A.3**.
>
> ***reference:***
>
> [1]: Brandes N, Ofer D, Peleg Y, et al. ProteinBERT: a universal deep-learning model of protein sequence and function[J]. Bioinformatics, 2022, 38(8): 2102-2110.
>
> [2]: Elnaggar A, Heinzinger M, Dallago C, et al. Prottrans: Toward understanding the language of life through self-supervised learning[J]. IEEE transactions on pattern analysis and machine intelligence, 2021, 44(10): 7112-7127.

---

> ### Author Response · Authors · 2024-11-22
>
> ### Question 2 (comparison missing)
>
> **Q2:** *In Table 2, as you described in Section 4.2 “Furthermore, as illustrated in Tab 2 comparisons between the 8M-cleaned and PB-cleaned version reveal that, pretrained models consistently outperform their from-scratch counterparts”, whether different parameters size cause this phenomenon？, we find that lacking the ProtAC-ESM2-35M-cleaned models, why don’t you compare their performance？You could include results for the in ProtAC-ESM2-35M-cleaned models.*
>
> **A2:**
>
> Thanks to this reviewer for raising questions about the model comparison results and providing suggestions for the completeness of the comparison!
>
> We believe that the number of parameters is not the main reason for the performance gap, because we deliberately selected two models with similar parameter numbers to avoid the possible differences caused by the number of parameters. The reviewer can view the details of ProtAC-ProteinBERT and ProtAC-ESM2-8M in **Tab S1**, and their parameter numbers only differ by 2M.
>
> We guess that the reviewer is referring to the supplementary ESM2-35M-uncleaned model, because the results of the ESM2-35M-cleaned model are already in Tab2. Therefore, **we supplemented the results of the ESM2-35M-uncleaned model in** **Tab2**.

---

> ### Author Response · Authors · 2024-11-22
>
> ### Question 3 (additional noise)
>
> **Q3:** *During the process of cleaning the large protein datasets, does this process cause additional noise to influence the quality of cleaned datasets. You are supposed to explain how to reduce additional noise, or quantify the quality/noise level of the cleaned datasets compared to the original.*
>
> **A3:**
>
> We sincerely appreciate this insightful question regarding potential noise introduction during our dataset cleaning process.
>
> Indeed, we acknowledge that intermediate rounds of cleaning may temporarily introduce prediction noise, as evidenced in Fig. 6a. Our analysis shows that in early rounds, the model occasionally predicts some incorrect GO terms. However, this observation precisely demonstrates the necessity and effectiveness of our recursive cleaning strategy. The key strength of our recursive approach lies in its self-correcting nature. Temporary inconsistencies are systematically eliminated through subsequent iterations. This convergence towards stable and accurate annotations validates the robustness of our recursive cleaning methodology. The final stable state of annotations, achieved after multiple rounds, provides strong empirical evidence for the quality of our cleaned dataset. We have provided more specific examples in **Tab. S5 and S9** which would be helpful in further illustrating this point.
>
> In order to quantify the noise level of the cleaned dataset, we introduced Jaccard Similarity. We use it to compare the distribution similarity between the cleaned dataset and the ground truth dataset. The higher the similarity, the closer the cleaned dataset is to the true annotation, that is, the lower the noise level. We applied it on the Swiss-caption dataset. The Jaccard similarity coefficient measures the overlap between two sets by dividing the size of their intersection by the size of their union, shown as the following equation:
> \begin{equation}
>     J(A,B) = \frac{|A \cap B|}{|A \cup B|} = \frac{\sum_{i=1}^n \min(A_i, B_i)}{\sum_{i=1}^n \max(A_i, B_i)}
> \end{equation}
> Here $A$ denotes the cleaned dataset, $B$ denotes Swiss-caption, and the results are shown in Table below (**Tab. S10** in the revised manuscript). As the number of cleaning rounds continues, the similarity increases, which means that the noise level is decreasing.
>
> | Model           | Round | jaccard similarity |
> | --------------- | ----- | ------------------ |
> | ProtAC-ESM2-35M | 1     | 0.2425             |
> |                 | 2     | 0.2862             |
> |                 | 3     | 0.2821             |
> |                 | 4     | 0.2820             |
> | ProtAC-ESM2-8M  | 1     | 0.1622             |
> |                 | 2     | 0.2350             |
> |                 | 3     | 0.2363             |
> |                 | 4     | 0.2301             |
> | ProtAC-PB       | 1     | 0.1011             |
> |                 | 2     | 0.1475             |
> |                 | 3     | 0.1594             |
> |                 | 4     | 0.1744             |
>
> We also introduced adaptive training in pertaining (see **Appendix B.2.6**) to enhance the training efficiency and counted the number of samples updated in training steps (see **Fig. S7**). As cleaning rounds continue, the number of updated samples in the training step gradually decreases, and it can be seen that the noise level of the pretraining dataset is reduced.

---

> ### Author Response · Authors · 2024-11-22
>
> ### Question 4 (open source)
>
> **Q4:** *In addition, authors should provide access to code and datasets so that other researchers can reproduce the results.*
>
> **A4:**
>
> Thanks to the reviewer for the request to open source the code. We are currently organizing the code and dataset. We promise to release the code, model weights, and cleaned dataset as soon as possible!

---

> ### Author Response · Authors · 2024-11-27
> **Request for Follow-up on Rebuttal Response**
>
> Dear Reviewer S9Fj,
>
> We noticed that **the deadline for submitting our revised PDF is approaching**, and we have not yet received your feedback on our rebuttal. Your expertise and insights are invaluable to us, and your comments would greatly help improve the quality of our work.
>
> **We would be very grateful if you could share your thoughts on our responses at your earliest convenience.** Your dedication to the review process is deeply appreciated.
>
> Thank you again for your time and effort.
>
> Best regards,
>
> Authors

---

### Official Review · Reviewer_HnpA · 2024-11-03

**Soundness:** 2
**Presentation:** 3
**Contribution:** 3
**Rating:** 5
**Confidence:** 5

**Summary:**

This paper presents ProtAC, an automatic cleaning framework for large protein datasets using multimodal learning with sequence and functional information. The core SAM module ensures proper sequence-annotation matching. ProtAC follows a cyclic process of pretraining, finetuning, and cleaning, progressively improving protein function prediction.

**Strengths:**

The proposed multi-round training strategy allows the model to progressively clean noisy datasets by predicting and selecting credible protein functional information, leading to mutual improvement in both dataset quality and model performance.

**Weaknesses:**

One concern with this paper is the unclear source of the noisy data. Is the noise stemming from the functional labels (e.g., GO annotations) or from unreliable textual information? GO annotations are typically well-supported by literature, so the rationale for annotation errors is not sufficiently explained. Lines 88-99 lack citations to support the claim of noisy data, making the subsequent discussion difficult to follow.

Additionally, the paper's characterization of OntoProtein is inaccurate. OntoProtein leverages biomedical text by incorporating it into a knowledge graph, contrary to the authors' description that it does not utilize biomedical text-related information.

The paper should also address why its results do not outperform knowledge graph-based methods like OntoProtein. Furthermore, it would be beneficial to compare ProtAC with models such as ProtST, which also utilizes protein sequences and biomedical texts, and ProtT3, a protein-text generation method, both of which could provide meaningful points of comparison for representation learning.

**Questions:**

Nan

---

> ### Author Response · Authors · 2024-11-22
>
> ### Weakness 1 (noise claim)
>
> **W1:** *One concern with this paper is the unclear source of the noisy data. Is the noise stemming from the functional labels (e.g., GO annotations) or from unreliable textual information? GO annotations are typically well-supported by literature, so the rationale for annotation errors is not sufficiently explained. Lines 88-99 lack citations to support the claim of noisy data, making the subsequent discussion difficult to follow.*
>
> **A1:**
>
> The noise originates from the methodology employed by UniProt to annotate protein sequences with GO annotations. According to the UniProt official documentation [https://www.uniprot.org/help/evidences#evidence-types-used-for-go-annotations], these annotations are derived from different methods with varying levels of confidence. The evidence types include experimental evidence codes, phylogenetically-inferred annotations, and computational analysis, among others. Variations in testing methods, experimental conditions, and analytical approaches for different proteins result in corresponding differences in their functional annotation processes. In addition, sequences in UniProt are categorized into two types: manually annotated (Swiss-Prot) and computationally analyzed (UniProtKB/TrEMBL), which also contributes to the variability in the quality of functional annotations across different sequences [1].
>
> In addition, we have included the web links [https://www.uniprot.org/help/evidences] [https://www.ebi.ac.uk/QuickGO/term/ECO:0007669] and relevant references [1][2] regarding the automatic annotation methods used by UniProt and GO annotations in the revised manuscript. This clarification explicitly addresses the specific reasons behind the noise in the functional annotations of the UniRef dataset, thereby improving the logical flow of the article.
>
> ***reference:***
>
> [1]: MacDougall A, Volynkin V, Saidi R, et al. UniRule: a unified rule resource for automatic annotation in the UniProt Knowledgebase[J]. Bioinformatics, 2020, 36(17): 4643-4648.
>
> [2]: Aleksander S A, Balhoff J, Carbon S, et al. The gene ontology knowledgebase in 2023[J]. Genetics, 2023, 224(1): iyad031.

---

> ### Author Response · Authors · 2024-11-22
>
> ### Weakness 2 (description inaccuracy)
>
> **W2:** *Additionally, the paper's characterization of OntoProtein is inaccurate. OntoProtein leverages biomedical text by incorporating it into a knowledge graph, contrary to the authors' description that it does not utilize biomedical text-related information.*
>
> **A2:**
>
> We sincerely thank the reviewer for pointing out our inaccurate characterization of OntoProtein. We have revised our description to correctly acknowledge that OntoProtein does utilize biomedical text information through its knowledge graph integration. The relevant section (**Section. 2 Related Work: Protein Multimodal Learning**) in the paper has been updated to avoid any misrepresentation.

---

> ### Author Response · Authors · 2024-11-22
>
> ### Weakness 3 (more baseline)
>
> **W3:** *The paper should also address why its results do not outperform knowledge graph-based methods like OntoProtein. Furthermore, it would be beneficial to compare ProtAC with models such as ProtST, which also utilizes protein sequences and biomedical texts, and ProtT3, a protein-text generation method, both of which could provide meaningful points of comparison for representation learning.*
>
> **A3:**
>
> We acknowledge these important points regarding model comparisons:
>
> - For the reasons why our results do not outperform ontoprotein:
>   - OntoProtein's knowledge graph structure provides a rich, structured representation of protein-function relationships that explicitly captures hierarchical and semantic connections between functional annotations, while ProtAC relies primarily on direct sequence-annotation matching without such structured knowledge integration.
>   - OntoProtein employs a significantly larger model with more parameters than ProtAC (<100M parameters), allowing it to capture more complex patterns and relationships in the data. However, this comes at the cost of higher computational requirements and reduced accessibility compared to ProtAC's more efficient design.
> - While our current results may not surpass OntoProtein, ProtAC offers a complementary approach focused on data quality improvement rather than competing directly with knowledge graph based architectures on protein function prediction.
> - We agree that comparisons with ProtST and ProtT3 would provide valuable insights. We have expanded our evaluation to include:
>   - Adding ProtST as another important baseline in **Tab. 1**.
>   - Discussion of the difference between ProtT3 and other performant PLMs in **Appendix A.3**, since it's for protein text generation, not directly for protein function prediction tasks like GO, EC and KW.
>
> These additional comparisons would help better position ProtAC within the broader landscape of protein representation learning methods.

---

> ### Author Response · Authors · 2024-11-27
> **Request for Follow-up on Rebuttal Response**
>
> Dear Reviewer HnpA,
>
> We noticed that **the deadline for submitting our revised PDF is approaching**, and we have not yet received your feedback on our rebuttal. Your expertise and insights are invaluable to us, and your comments would greatly help improve the quality of our work.
>
> **We would be very grateful if you could share your thoughts on our responses at your earliest convenience.** Your dedication to the review process is deeply appreciated.
>
> Thank you again for your time and effort.
>
> Best regards,
>
> Authors

---

### Official Review · Reviewer_nKxg · 2024-11-04

**Soundness:** 2
**Presentation:** 2
**Contribution:** 2
**Rating:** 5
**Confidence:** 4

**Summary:**

The paper introduces a novel approach leveraging knowledge distillation techniques for cleaning large protein datasets, aimed at improving protein function prediction. The authors propose a multimodal learning model, ProtAC, that incorporates sequence and functional annotations to enhance dataset quality and model training through an iterative train-finetune-clean cycle.

**Strengths:**

1. Applying knowledge distillation to clean large protein datasets is a novel and promising approach.
2. The resulting pre-trained model shows improvement over the baseline model, ESM2, suggesting potential in the proposed method.

**Weaknesses:**

1. Importance "Performant PLM Baselines" like ProtST is missing. Given ProtST outperforms OntoProtein by a large margin, the performance in Tab. 1 remains to be justified. This omission also raises questions about the comparative effectiveness of the proposed approach. Also see Questions
2. The reliability of the automatically annotated dataset is mainly accessed by case study, which may lead to doubt in applying the dataset further in future.

**Questions:**

1. High-performing baselines like ProtST is missing for experiments.
2. In Fig. 4 and Tab. 2, ProtAC-ESM2-35M-Uncleaned is missing.  Can authors provide these missing comparisons?
3. For Fig. 4d, the comparison may be unfair due to different computational resources used. For Scratch-epoch4 and Scratch-original, the pre-training only lasts for 1 epoch. But for ProtAC-ProteinBERT-cleaned, it experienced 4 epochs staged pre-training.
4. Could authors further explain "Among the 20 sampled clusters, clusters are no longer present in UniRef50"? How does this relate to the prediction of transmembrane regions with the Phobius tool?
5. In ablation, why does removing MLM loss lead to better performance for KW prediction?

---

> ### Author Response · Authors · 2024-11-22
>
> ### Weakness 1 & Question 1 (baseline missing)
>
> **W1:** *Importance "Performant PLM Baselines" like ProtST is missing. Given ProtST outperforms OntoProtein by a large margin, the performance in Tab. 1 remains to be justified. This omission also raises questions about the comparative effectiveness of the proposed approach. Also see Questions*
>
> **Q1:** *High-performing baselines like ProtST is missing for experiments.*
>
> **A1:**
>
> Thank you for this valuable feedback regarding ProtST. We have carefully considered your comments and would like to address them as follows:
>
> In response to your concern, we have now included ProtST's performance results in **Tab. 1** of our revised manuscript for a more comprehensive comparison.
>
> Regarding our previous omission of ProtST, we would like to offer several considerations:
>
> 1. While both approaches address protein function prediction, there are fundamental differences in the input modalities. ProtST leverages textual information as one of its multiple modalities, whereas our approach utilizes a simpler one-hot encoded GO term representation. This architectural difference initially led us to focus on more directly comparable baselines.
> 2. It is worth noting that ProtST employs a substantially larger parameter count compared to our model (**nearly 10 times**). While we acknowledge ProtST's impressive performance, the significant disparity in model size poses challenges for fair comparison under equivalent computational constraints.
> 3. We did consider adopting ProtST as one of our backbone architectures. However, due to limited computational resources at our disposal, we had to make the practical decision to focus on more resource-efficient alternatives.

---

> ### Author Response · Authors · 2024-11-22
>
> ### Weakness 2 (case study)
>
> **W2:** *The reliability of the automatically annotated dataset is mainly accessed by case study, which may lead to doubt in applying the dataset further in future.*
>
> **A2:**
>
> We thank the reviewer for the suggestion to provide a more comprehensive validation of the dataset automatical annotation. Given the current computational capabilities, the results presented in our manuscript provide a reasonable validation for the large-scale protein sequence functional annotation/cleaning task.
>
> 1. GO annotation prediction is one of the most common tasks in protein sequence functional annotation and has been used in the Critical Assessment of Functional Annotation (CAFA) to evaluate the performance of various protein function prediction methods. In this study, the task is employed to assess the model architecture and data cleaning strategies, with comparisons made to baseline approaches, resulting in SOTA performance. This provides compelling evidence for the effectiveness of ProtAC in large-scale data cleaning.
> 2. The complexity of functional annotation in biological data is the primary reason why the number of high-quality functional annotations is currently much lower than the number of protein sequences. Furthermore, due to variations in the methods of functional validation, the reliability of different annotations varies significantly. Our biological case study involved manual review based on randomly sampled cleaned data, following a process similar to the manual curation workflow outlined by UniProt. Importantly, no data with different levels of cleaning quality were selectively chosen during this process, making this one of the most reliable validation methods available, aside from wet lab experiments. Similarly, in the ProtNLM model [https://www.uniprot.org/help/ProtNLM] used by UniProt for protein function and name annotation, validation of annotation performance was also conducted through manual review.
> 3. In **Fig. S2 and S3**, we demonstrated the cleaning effect of ProtAC on large-scale data by performing dimensionality reduction visualization of sequences with specific GO annotations before and after data cleaning.
> 4. We additionally launched another more lagre-scale investigation showed in **Tab. S9**, where we have included the number of manual reviews of the GO annotations for the same UniRef clusters after each round of data cleaning, to further demonstrate the reliability of the automatically annotated dataset.
> 5. We added a general experiment on Swiss-caption dataset to compare the distribution similarity between the cleaned dataset and the ground truth (quantified using Jaccard Similarity, a statistical method) (see **Appendix B.3.4 and Tab. S10** for detailed results, or the table showed below). The results show that the noise level of our cleaned dataset is gradually decreasing.
>
> | Model           | Round | jaccard similarity |
> | --------------- | ----- | ------------------ |
> | ProtAC-ESM2-35M | 1     | 0.2425             |
> |                 | 2     | 0.2862             |
> |                 | 3     | 0.2821             |
> |                 | 4     | 0.2820             |
> | ProtAC-ESM2-8M  | 1     | 0.1622             |
> |                 | 2     | 0.2350             |
> |                 | 3     | 0.2363             |
> |                 | 4     | 0.2301             |
> | ProtAC-PB       | 1     | 0.1011             |
> |                 | 2     | 0.1475             |
> |                 | 3     | 0.1594             |
> |                 | 4     | 0.1744             |

---

> ### Author Response · Authors · 2024-11-22
>
> ### Question 2 (comparison missing)
>
> **Q2:** *In Fig. 4 and Tab. 2, ProtAC-ESM2-35M-Uncleaned is missing. Can authors provide these missing comparisons?*
>
> **A2:**
>
> Thanks to this reviewer for his great suggestion on the completeness of the experimental comparison! We have added the keyword prediction results of the 35M-uncleaned model in **Tab. 2** and the GO prediction results of the 35M-uncleaned model in **Fig. 4**. The reviewer can check them together in the revised manuscript!

---

> ### Author Response · Authors · 2024-11-22
>
> ### Question 3 (unfair comparison)
>
> **Q3:** *For Fig. 4d, the comparison may be unfair due to different computational resources used. For Scratch-epoch4 and Scratch-original, the pre-training only lasts for 1 epoch. But for ProtAC-ProteinBERT-cleaned, it experienced 4 epochs staged pre-training.*
>
> **A3:**
>
> We sincerely appreciate your careful attention to the experimental comparisons in Fig. 4d. We would like to clarify several important points:
>
> First, we should note that **all models shown in Fig. 4d are variants of ProtAC-ESM2-8M, rather than ProtAC-ProteinBERT-cleaned** as mentioned in your comment.
>
> Regarding the experimental design, our intention was to conduct two separate comparisons:
>
> 1. A direct comparison between "Scratch-epoch4" and "Scratch-original"to demonstrate the effectiveness of our cleaned dataset
> 2. A separate comparison between models with multiple training epochs ("Cleaned" and "Uncleaned") to validate our cleaning strategy
>
> We did not intend to compare models across these two groups due to their different training configurations. We acknowledge that presenting all results in a single figure might have caused some confusion. To address this, **we have updated Fig. 4d in our revised manuscript by adding an orange dashed line to clearly separate these two distinct comparison groups**. We have **also added a clarifying note in the figure caption** explaining that the dashed line demarcates two separate comparative analyses.
>
> We apologize for any confusion our presentation may have caused and hope these clarifications address your concerns. Please let us know if you need any additional information or clarification.

---

> ### Author Response · Authors · 2024-11-22
>
> ### Question 4 (unclear biological explanation)
>
> **Q4:** *Could authors further explain "Among the 20 sampled clusters, clusters are no longer present in UniRef50"? How does this relate to the prediction of transmembrane regions with the Phobius tool?*
>
> **A4:**
>
> We thank the reviewer for this question of explaination for the case study. UniProt is updated roughly every eight weeks. For UniRef, certain clusters may be deleted or reassigned to other clusters as the dataset is updated. As a result, some clusters from older versions may no longer be present in the latest release of UniRef. The statement "*Among the 20 sampled clusters, clusters are no longer present in UniRef50*" refers to this situation. Therefore, the changes in clusters due to database updates are unrelated to the prediction of transmembrane regions using the Phobius tool.

---

> ### Author Response · Authors · 2024-11-22
>
> ### Question 5 (ablation study)
>
> **Q5:** *In ablation, why does removing MLM loss lead to better performance for KW prediction?*
>
> **A5:**
>
> Thank you to the reviewer for the question about the performance of the module ablation study. In fact, we often find this phenomenon in other people's multimodal PLM work. For example, in Table 15 of ProtST [1] (one of the SOTA PLMs in protein function prediction), the author did an ablation study of pre-training losses on function annotation. We found that removing the MLM loss actually improved the accuracy of some prediction tasks. Here, for our work, we give a possible speculation: KW prediction is annotation prediction, so it is strongly related to annotation prediction loss, and MLM and SAM loss will dilute the weight of loss, so removing them can allow the model to focus on anno prediction, but this does not help seq-anno matching, so they cannot be removed. We hope that this answer can answer the reviewer's doubts, and any further questions are welcome!
>
> We appreciate your rigorous review and hope these clarifications address your concerns. We would be happy to provide any additional information or clarification you may need.
>
> ***reference:***
>
> [1]: Xu M, Yuan X, Miret S, et al. Protst: Multi-modality learning of protein sequences and biomedical texts[C]//International Conference on Machine Learning. PMLR, 2023: 38749-38767.

---

> ### Author Response · Authors · 2024-11-27
> **Request for Follow-up on Rebuttal Response**
>
> Dear Reviewer nKxg,
>
> We noticed that **the deadline for submitting our revised PDF is approaching**, and we have not yet received your feedback on our rebuttal. Your expertise and insights are invaluable to us, and your comments would greatly help improve the quality of our work.
>
> **We would be very grateful if you could share your thoughts on our responses at your earliest convenience.** Your dedication to the review process is deeply appreciated.
>
> Thank you again for your time and effort.
>
> Best regards,
>
> Authors

---

### Official Review · Reviewer_Yxmh · 2024-11-04

**Soundness:** 2
**Presentation:** 3
**Contribution:** 2
**Rating:** 5
**Confidence:** 4

**Summary:**

The authors propose a novel method to recursively clean the large-scale protein dataset, namely UniRef50 and Swiss-Prot in the paper's context. The authors develop a multi-step cleaning procedure alone with a joint model for predicting the annotation as well as judging whether the annotation is correct. Experiments show the efficiency of the proposed methodology.

**Strengths:**

1. The manuscript is generally well-written, along with good figures that can clearly explain the details.
2. The analysis authors conduct are solid and the topic of cleaning protein datasets can be helpful to the wide community of AI4protein.

**Weaknesses:**

1.It is a solid idea to clean the existing protein datasets and strengthen their quality. However, the data cleaning procedure is not novel since this is a well-explored topic in natural language processing. So in my opinion, the major contribution of the paper is on the cleaned dataset itself, not on the methodology, while the main body of the paper is devoted to it.

2.The authors specifically emphasize on the efficiency of their methodology. But efficiency should not be the primary concern since you only need to produce the cleaned dataset once. The performance should be dominantly important instead. However, the large gap between esm2 and protbert seems to show that the performance mainly relies on the pretrained language model, not on the procedure itself.

**Questions:**

1. The figure4, 5 and 6 have too small fonts. Please make them larger.

---

> ### Author Response · Authors · 2024-11-22
>
> ### Weakness 1 (paper focus)
>
> **W1:** *It is a solid idea to clean the existing protein datasets and strengthen their quality. However, the data cleaning procedure is not novel since this is a well-explored topic in natural language processing. So in my opinion, the major contribution of the paper is on the cleaned dataset itself, not on the methodoloy, while the main body of the paper is devoted to it.*
>
> **A1:**
>
> Many thanks to reviewer Yxmh for reviewing our paper and recognising the significance of our work to the field of AI for protein!
>
> In general, to our knowledge, this work is the first to propose the idea of cleaning a protein dataset and to design and validate a complete and efficient pipeline.
>
> First of all, we strongly agree with the reviewer that **the cleaned dataset should be the focus of the paper, and in fact this has been one of the main contributions of our paper**: we cleaned a cleaned dataset and performed a series of rigorous experiments on it to verify the reliability of the dataset itself and the validity of the cleaning process. For example: In Sec4.2, **Fig.4b** shows the results of pretraining the same model using cleaned and original dataset respectively, showing that the model pretrained with cleaned dataset is faster and eventually achieves higher function prediction capability; **Fig.4d** visualises the final function prediction capability of the same model after four rounds of training, in which the model trained on cleaned dataset performs significantly better; these are all validations around the dataset itself, and all illustrate that the use of cleaned dataset can be used to predict the function of the same model after four rounds of training, all of which also illustrate the advantages of using cleaned dataset for model pretraining. In Sec4.3, we demonstrate that the model's understanding of protein function is reliable by exploring the changes in GO annotations of the same protein over multiple rounds of cleaning (**Fig. 6a**) and by querying the dataset before and after cleaning for proteins that have a particular functional annotation (transmembrane) (**Fig. 6b**) which also demonstrates that our cleaned dataset has more plausible functional annotations than the original dataset.
>
> In addition, to strengthen our focus on dataset itself, **we supplement the appendix with new biological analysis experiments**, which test the functional description of cleaned dataset at a more general level, demonstrating the improvement of the quality of large-scale protein datasets by our cleaning strategy. We conducted manual review of the GO annotations for additional UniRef clusters. As shown in **Tab. S9**, the majority of newly added GO annotations for the UniRef50 clusters analyzed in each cleaning round are supported either by their presence in the latest version of the UniRef50 database or by evidence from family databases such as InterPro and Pfam.
>
> Second, we respect the reviewer's opinion but disagree with the reviewer on the importance of the methodology. We would like to emphasise that how to clean the data is very important, and what we propose is not only cleaning but also a training pipeline, which can accelerate the training of the model. We also disagree with the reviewer that the data cleaning procedure is not novel, as we have innovatively designed the modules in the recursive cleaning workflow to address the multimodal nature of proteins, which cannot be directly appropriated from the models and methods in the field of natural language processing. In addition, we believe that validation of the proposed methodology is a necessary process for the work, and thus both the efficiency of the methodology and the performance demonstration are necessary, which greatly contributes to the completeness of the article.
>
> Finally, thanks again to reviewer Yxmh for the above suggestions on our work, and any more detailed suggestions on how to evaluate cleaned dataset itself are very welcome!

---

> ### Author Response · Authors · 2024-11-22
>
> ### Weakness 2 (methodology performance)
>
> **W2:** *The authors specifically emphasize on the efficiency of their methodology. But efficiency should not be the primary concern since you only need to produce the cleaned dataset once. The performance should be dominantly important instead. However, the large gap between esm2 and protbert seems to show that the performance mainly relies on the pretrained language model, not on the procedure itself.*
>
> **A2:**
>
> Many thanks to reviewer Yxmh for his thoughts on efficiency and performance in methodology!
>
> First of all, we are afraid that we disagree with the reviewer on ‘*efficiency should not be the primary concern since you only need to produce the cleaned dataset once*’. We must emphasise again that **our proposed pipeline is one of our main focuses, which combines the cleaning of the dataset with the training of the model.** Protein datasets are diverse, with varying sizes and cleaning complexities. To enhance the transferability of ProtAC across different datasets and cleaning tasks, it is essential to consider efficiency,  as it will be used repeatedly. The idea of cycle cleaning is not only limited to the protein domain, but can even be extended to domains related to Natural Language Processing, since to the best of our knowledge this idea has been proposed for the first time.
>
> Secondly, in response to reviewer Yxmh's statement that ‘*the large gap between esm2 and protbert seems to show that the performance mainly relies on the pretrained language model, not on the procedure itself*’, we acknowledge that there are gaps between the pretrained models themselves, but that is not the point we are trying to convey by showing performance. We focus on comparing the huge performance gains that can be made by models adopting our workflow.
>
> | Method              |       GO-BP       |       GO-BP       |       GO-MF       |       GO-MF       |       GO-CC       |       GO-CC       |        EC         |        EC         |
> | :------------------ | :---------------: | :---------------: | :---------------: | :---------------: | :---------------: | :---------------: | :---------------: | :---------------: |
> |                     |     **AUPR**      |     **Fmax**      |     **AUPR**      |     **Fmax**      |     **AUPR**      |     **Fmax**      |     **AUPR**      |     **Fmax**      |
> | ESM2-8M             |       0.154       |       0.284       |       0.410       |       0.394       |       0.187       |       0.373       |       0.477       |       0.468       |
> | **ProtAC-ESM2-8M**  | **0.239(+55.2%)** | **0.354(+24.6%)** | **0.454(+10.7%)** | **0.423(+7.4%)**  | **0.307(+64.2%)** | **0.431(+15.5%)** | **0.579(+21.4%)** | **0.558(+19.2%)** |
> | ESM2-35M            |       0.212       |       0.340       |       0.501       |       0.489       |       0.248       |       0.417       |       0.562       |       0.571       |
> | **ProtAC-ESM2-35M** | **0.268(+26.4%)** | **0.379(+11.5%)** | **0.577(+15.2%)** | **0.603(+23.3%)** | **0.321(+29.4%)** | **0.461(+10.6%)** | **0.615(+9.4%)**  | **0.619(+8.4%)**  |
>
> We agree with the reviewer that ‘*the performance should be dominantly important*’ and our work aims to demonstrate this, as can be seen in the table above, where two different parameterised versions of esm2 are trained using our workflow, both versions of esm2 with different parameters have significantly improved protein function prediction using our training strategy, which proves that our workflow is very effective. We have added this table and its description to the **Appendix B.2.5 and Tab. S6**.
>
> Finally, thanks again to reviewer Yxmh for the comments on the performance of our work, and any more in-depth discussion is welcome!

---

> > ### Comment · Reviewer_Yxmh · 2024-12-02
> > **I have updated the scores**
> >
> > Hello dear authors, you have done sufficient work to persuade me, especially your additional results shown against weakness 2. I have raised my rating to 5. Good luck!

---

> ### Author Response · Authors · 2024-11-22
>
> ### Question 1 (figure visuals)
>
> **Q1:** *The figure4, 5 and 6 have too small fonts. Please make them larger.*
>
> **A1:**
>
> Thanks to this reviewer's suggestion to improve the figures in the article, we have adjusted the fonts in Fig. 4 to 6 according to the suggestion, please check them in the latest version of the manuscript!
>
> ### Summary of response to reviewer Yxmh's comments
>
> The reviewer provided constructive feedback on our paper focus, methodology performance, and figure visuals, and we have made the appropriate experimental additions, textual refinements, and graphical adjustments to the article based on the suggestions. We thank the reviewer for his contribution to the completeness and clarity of our work!

---

> ### Author Response · Authors · 2024-11-27
> **Request for Follow-up on Rebuttal Response**
>
> Dear Reviewer Yxmh,
>
> We noticed that **the deadline for submitting our revised PDF is approaching**, and we have not yet received your feedback on our rebuttal. Your expertise and insights are invaluable to us, and your comments would greatly help improve the quality of our work.
>
> **We would be very grateful if you could share your thoughts on our responses at your earliest convenience.** Your dedication to the review process is deeply appreciated.
>
> Thank you again for your time and effort.
>
> Best regards,
>
> Authors

---

### Author Response · Authors · 2024-11-24

## Overall Response to Reviewers

We sincerely thank all reviewers for their thoughtful and constructive feedback. Based on the reviews, we have made substantial improvements to our manuscript through additional experiments, analyses, and clarifications. Here are the key changes and responses:

### Dataset Quality and Validation

- Added comprehensive validation of the automatically annotated dataset through:
  - Large-scale manual review of GO annotations across cleaning rounds **(Tab. S9)**
  - New statistical analysis using Jaccard Similarity to quantify distribution similarity between cleaned dataset and ground truth **(Tab. S10)**

### Extended Comparisons and Baselines

- Added ProtST, SaProt as a crucial baseline in performance comparisons **(Tab. 1)**
- Included ProtAC-ESM2-35M-Uncleaned results in **Fig. 4 and Tab. 2**
- Clarified performance comparisons in **Fig. 4** with improved visualization and explanations
- Added discussion of ProtT3 and other recent protein language models in **Appendix A.3**

### Methodology and Performance

- Demonstrated significant performance improvements across different model architectures:
  - ESM2-8M: improvements of 10.7-64.2% across metrics
  - ESM2-35M: improvements of 9.4-29.4% across metrics
- Clarified the importance of both efficiency and performance in our pipeline
- Added more comprehensive biological analysis experiments in the **Appendix B.3**

### Technical Clarity

- Provided clearer explanations of data noise sources with proper citations
- Corrected characterization of OntoProtein's use of biomedical text
- Improved figure readability by adjusting font sizes (**Fig. 4-6**)
- Added detailed explanations for ablation study results

### Documentation Updates

- Added relevant citations and references regarding UniProt annotation methods
- Supplementary description of the recursive cleaning workflow in Appendix
- Enhanced figure captions and explanations throughout

Other additional revisions please see the newest version of our manuscript!

The additional experiments and analyses have strengthened our validation of the cleaned dataset's quality while maintaining our paper's dual focus on both the methodology and the resulting cleaned dataset. We believe these changes have substantially improved the paper's completeness and clarity.

---

### Meta-Review · Area_Chair_t7XB · 2024-12-20

**Metareview:**

The paper received five reviews with ratings of 5, 5, 5, 6, and 5. The reviewers raised several major concerns about the paper, including the lack of comparisons with key baseline methods, insufficient support for the claim of noisy data, inaccurate characterization of OntoProtein, questionable scalability, limited generalizability and effectiveness due to the reliance on extensive computations, a small and manually annotated dataset, and restricted model parameters. Based on the majority of reviewers' opinions, this paper is recommended for rejection.

**Additional Comments On Reviewer Discussion:**

Two reviewers engaged in discussions with the authors, with one reviewer raising their rating score to 5.

---

### Decision · Program_Chairs · 2025-01-22

Reject